# GRACE: TOWARDS REALISTIC MULTIMODAL SINGLE-CELL DATA MATCHING

## ABSTRACT

Single-cell multi-omics technologies (e.g., scRNA-seq and scATAC-seq data) have provided more comprehensive insights for understanding cellular conditions and activities in recent years. However, multimodal representation learning for transcriptomics data remains a challenging problem due to heterogeneous relationships and label scarcity in reality. In this work, we propose a novel approach named Geometric Relation Exploration with Cross-modal Supervision (GRACE) for realistic multimodal single-cell matching. In particular, we map both multimodal data into a shared embedding space by maximizing the log-likelihood of ZINB distributions. To reduce the semantic gap between multimodal data, we construct a geometric graph using mutual nearest neighbors to indicate cross-modal relations between samples for distribution alignment. Furthermore, to extract most pairwise information, we explore high-order relations in the geometric graph, which would be incorporated into a meta-learning paradigm for robust optimization. In addition, to further mitigate label scarcity, we introduce a nonparametric way to generate label vectors for unlabeled data for cross-modal supervision across different modalities. Extensive experiments on several benchmark datasets validate the superiority of the proposed GRACE compared to various baselines. In general, compared to the second-best method, GRACE exhibits an average performance improvement of 6.71% and 14.17% for the R2A task and A2R task, respectively. Code is available at `https://anonymous.4open.science/r/GRACE`.

## 1 INTRODUCTION

Modern single-cell multi-omics technologies (Chappell et al., 2018; Wen et al., 2022b) have made extensive achievements, which enable the measurement of cells from various modalities for understanding cell biology in health and disease. Among them, single-cell RNA-sequencing (scRNA-seq) (Kolodziejczyk et al., 2015), single-cell ATAC-sequencing (scATAC-seq) (Pott & Lieb, 2015), and single-cell DNA methylome sequencing (Karemaker & Vermeulen, 2018) quantify the gene expression, chromatin accessibility and DNA methylation of individual cells, respectively. There have also been sequencing technologies for the joint measurement of multi-omics information from the same cell such as CITE-seq (Stoeckius et al., 2017) and ASAP-seq (Mimitou et al., 2021). To understand and integrate data from various modalities, it is highly anticipated to develop a unified cell representation learning framework, which maps multimodal data to a common embedding space while preserving the original semantic relationships.

In literature, several multimodal single-cell data integration approaches are proposed (Lin et al., 2022; Li et al., 2023). Despite their tremendous progress, these single-cell multimodal integration approaches require a large number of labeled single-cell data (Wang et al., 2021; Huang et al., 2021) to boost the performance. Nevertheless, multimodal single-cell data frequently originate from different sources (Kiselev et al., 2019), where cell type information is not provided for all the data. In reality, there always exists a large number of economic unlabeled multimodal data (Qi et al., 2020). Moreover, the existing multimodal single-cell data integration approaches focus on transferring knowledge about cell types, while ignoring the more challenging problem of matching single-cell multimodal representations. This motivates us to study an underexplored problem of realistic multimodal single-cell data matching, which learns unified cell representations with cellular semantics incorporated by jointly using both labeled and unlabeled multimodal data.

In practice, formalizing a framework for realistic multimodal single-cell data matching remains challenging since two questions are required to tackle : (1) *How to obtain modality-invariant representations for multimodal single-cell data?* Note that scRNA-seq and scATAC-seq data offer different perspectives on cell-level descriptions (Green et al., 2022). Therefore, the distribution discrepancy in the hidden space (Andonian et al., 2022; Liu et al., 2021; Patel et al., 2023) usually leads to the semantic gap between the two modalities. (2) *How to learn discriminative cell representations under label scarcity?* Due to the absence of annotation information, cell representations of unlabeled data could be of poor quality without proper semantics incorporated (Li et al., 2020). This leads to insufficient supervision during the learning process. In addition, the heterogeneous relationship across multimodal data makes the problem more complicated.

To address these issues, we propose a new approach named Geometric Relation Exploration with Cross-modal Supervision (GRACE). Our GRACE utilizes separate auto-encoders to transform multimodal high-dimensional single-cell data into a unified embedding space. To preserve the most information, we reconstruct the original count data with likelihood maximization by incorporating underlying zero-inflated negative binomial (ZINB) (Clivio et al., 2019) distributions. The core of our GRACE is to explore hierarchical geometric relations between unlabeled multimodal samples. In particular, we construct a geometric graph among unlabeled samples by employing mutual nearest neighbors in the hidden space to illustrate the distribution discrepancy across modalities, which is minimized with the help of a memory bank. Furthermore, we investigate high-order relations in the geometric graph for extra supervision, which is integrated into a meta-learning framework (Vanschoren, 2018) for robust optimization. To mitigate label scarcity, we present a nonparametric strategy for generating label distributions by comparing unlabeled cell representations with support representations. These label distributions would be refined for informative signals for effective discriminative learning across modalities. Extensive experiments on a range of benchmark datasets validate the superiority of the proposed GRACE in comparison to competing baselines. The contribution of this work can be summarized as follows:

- *Problem Formulation.* We study an underexplored problem of realistic multimodal single-cell data matching, which extends multimodal learning into biological data understanding.

- *Novel Methodology.* On the one hand, GRACE explores hierarchical geometric relations among cross-modal unlabeled samples, which are incorporated into a meta-learning paradigm to ensure robust distribution alignment. On the other hand, GRACE introduces a nonparametric manner to generate label vector distributions for discriminative learning across different modalities.

- *Multifaceted Experiments.* Extensive experiments on a range of benchmark datasets validate the superiority of the proposed GRACE compared with diverse baseline methods in different settings.

## 2 BACKGROUND

**Prior Works.** Early efforts often focus on matrix factorization (Duren et al., 2018; Jin et al., 2020; Stein-O'Brien et al., 2018) and statistical models (Shen et al., 2009; Stuart et al., 2019; Welch et al., 2017). Matrix factorization (Wang & Zhang, 2012; Xu et al., 2020) is a powerful tool for dimension reduction, producing low-dimensional representations that facilitate cellular inference. In contrast, statistical models (Xiao et al., 2022) frequently employ intricate data distributions to characterize gene expression, followed by statistical inference with uncertainty. In recent years, deep learning has made significant strides in single-cell data integration (Tang et al., 2023). Some existing approaches utilize auto-encoders (Gong et al., 2021; Wu et al., 2021; Tu et al., 2022; Gala et al., 2019) to produce compact cell representations. Other approaches (Wen et al., 2022a; Wang et al., 2021) employ graph neural networks to model the relationships between genes and cells, and then utilize the message passing mechanism to generate discriminative representations. However, it's imperative to underscore that none of the existing works are suitable for our problem setting. They fall short in effectively leveraging both labeled and unlabeled multimodal data for efficient representation matching.

**Problem Definition.** To begin, we provide the problem definition of realistic multimodal single-cell data matching. Let $\mathcal{D}^{(1),w} = \mathcal{D}^{(1)} \cup \mathcal{D}^{(1),l}$ denote the scRNA-seq dataset with unlabeled data $\mathcal{D}^{(1)} = \{(\boldsymbol{x}_i^{(1)})\}_{i=1}^N$ and labeled data $\mathcal{D}^{(1),l} = \{(\boldsymbol{x}_i^{(1),l}, y_i^{(1),l})\}_{i=1}^{N^l}$. Let $\mathcal{D}^{(2),w} = \mathcal{D}^{(2)} \cup \mathcal{D}^{(2),l}$ denote the scATAC-seq dataset with unlabeled data $\mathcal{D}^{(2)} = \{(\boldsymbol{x}_i^{(2)})\}_{i=1}^N\}$ and labeled data $\mathcal{D}^{(2),l} = \{(\boldsymbol{x}_i^{(2),l}, y_i^{(2),l})\}_{i=1}^{N^l}$. $N^l$ and $N$ denote the number of labeled pairs and unlabeled pairs, respectively.

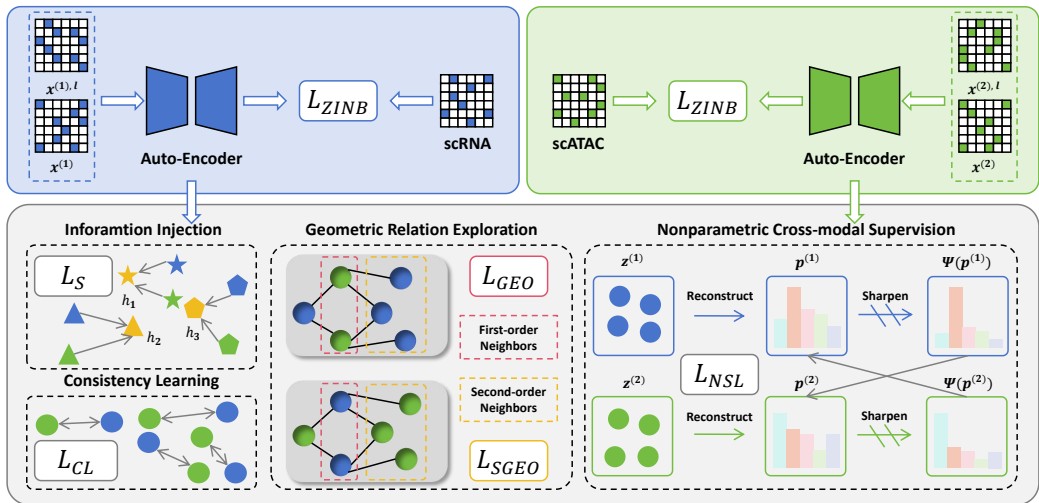

Figure 1: An overview of our GRACE. GRACE adopts separate auto-encoders to compress multimodal single-cell data into a common space. To reduce the semantic gap, we construct a geometric graph and explore first-order and second-order neighbors for pairwise distance minimization. We also reconstruct label vectors in a nonparametric way for cross-modal supervision.

We aim to develop a representation learning model, which maps multimodal data into a common embedding space. Here, it is anticipated that these samples with the same semantics would be close compared with those with different semantics. During evaluation, given a query from one modality, samples from the other modality are ranked according to similarity scores in the embedding space.

## 3 METHODLOGY

### 3.1 FRAMEWORK OVERVIEW

This work investigates the realistic problem of realistic multimodal single-cell data matching, which is challenging due to heterogeneous relationships between modalities and label scarcity in practice. In brief, we propose a new approach named GRACE for this problem, which mainly consists of three modules as follows: (1) *Joint Representation Learning*, which leverages separate auto-encoders to compass high-dimensional and sparse single-cell data from different modalities with likelihood maximization; (2) *Geometric Relation Exploration*, which builds a geometric graph using mutual nearest neighbors at different orders and then explore relations among the hierarchical neighbors for robust semantics alignment with a bi-level meta learning paradigm; (3) *Nonparametric Semi-supervised Learning*, which reconstructs label vector distributions by comparing unlabeled data with labeled data, providing supervision for cross-modal consistency. The overview of our GRACE can be found in Figure 1. Next, we elaborate on the details of each module in the proposed approach.

### 3.2 JOINT REPRESENTATION LEARNING WITH LIKELIHOOD MAXIMIZATION

To map both multimodal data into a common embedding space, we utilize two separate auto-encoders for samples from different modalities. We characterize these samples using an underlying zero-inflated negative binomial (ZINB) distribution (Clivio et al., 2019) and maximize the log-likelihood for reconstruction to keep the most information of representations. Moreover, the distance between labeled samples and their corresponding anchors is reduced for semantics injection.

In particular, two feed-forward networks (FFNs) $\phi_e^{(1)}(\cdot)$ and $\phi_e^{(2)}(\cdot)$ are introduced to generate cell representations:

$$\boldsymbol{z}_i^{(1)} = \phi_e^{(1)}(\boldsymbol{x}_i^{(1)}), \boldsymbol{z}_i^{(2)} = \phi_e^{(2)}(\boldsymbol{x}_i^{(2)}). \tag{1}$$

To characterize the distribution of count data, we adopt ZINB distribution with three parameters, i.e., the mean ($\mu$), the dispersion ($\theta$) and the probability of dropout $\pi$. The ZINB distribution for any

given sample from both modalities $\boldsymbol{x}$ can be written as:

$$\text{ZINB}\left(\boldsymbol{x} \mid \pi, \mu, \theta\right) = \pi \delta_0\left(\boldsymbol{x}\right) + (1-\pi)\text{NB}\left(\boldsymbol{x} \mid \mu, \theta\right), \tag{2}$$

$$\text{NB}\left(\boldsymbol{x} \mid \mu, \theta\right) = \frac{\Gamma\left(\boldsymbol{x}+\theta\right)}{\boldsymbol{x}!\Gamma(\theta)}\left(\frac{\theta}{\theta+\mu}\right)^{\theta}\left(\frac{\mu}{\theta+\mu}\right)^{\boldsymbol{x}}, \tag{3}$$

where $\Gamma(\cdot)$ denotes the Gamma distribution and $\text{NB}(\cdot)$ denotes the negative binomial distribution. $\delta_0(\cdot)$ denotes a Dirac delta function. Different from the classic auto-encoders, we involve three heads in each decoder to generate the above three parameters, i.e., $\mu$, $\theta$ and $\pi$ for likelihood maximization. In other words, the loss objective can be formalized into:

$$L_{\text{ZINB}} = -\sum_{\boldsymbol{x}\in\mathcal{D}^{(1),w}\cup\mathcal{D}^{(2),w}} \log\left(\text{ZINB}\left(\boldsymbol{x} \mid \pi, \mu, \theta\right)\right). \tag{4}$$

Compared with the standard regression loss accompanied by Gaussian distribution (Ng et al., 2011), our loss objective is more suitable for non-negative count single-cell data. To inject semantics information, we project labels into embedding space, resulting in $C$ learnable anchors, i.e., $\boldsymbol{h}_1, \cdots, \boldsymbol{h}_C$. Then, we enforce the representations of labeled samples to approach their corresponding anchors. In formulation,

$$L_S = \sum_{\boldsymbol{x}_i^{(1)}\in\mathcal{D}^{(1),l}} ||\boldsymbol{z}_i^{(1)} - \boldsymbol{h}_{y_i^{(1)}}||_2^2 + \sum_{\boldsymbol{x}_i^{(2)}\in\mathcal{D}^{(2),l}} ||\boldsymbol{z}_i^{(2)} - \boldsymbol{h}_{y_i^{(2)}}||_2^2. \tag{5}$$

By minimizing the distance between deep representations between shared anchors, we can align representations from both modalities with semantics incorporated. However, in realistic scenarios, labeled samples are usually scarce (Li et al., 2020). Therefore, high-quality data matching cannot be guaranteed by reconstruction and supervised learning. To achieve this goal, we are required to design effective modules to make use of a large number of unlabeled data.

### 3.3 GEOMETRIC RELATION EXPLORATION FOR SEMANTICS ALIGNMENT

One major challenge is the semantic gap between different modalities. To reduce this gap, we propose to explore the hierarchical geometric relations for extensive unlabeled samples. Due to our joint representation learning, samples with similar semantics tend to gather and the distance between cross-modal pairs indicates the potential distribution discrepancy. Therefore, we build a geometric graph using cross-modal mutual nearest neighbors and minimize the distance between connected pairs for distribution alignment. High-order relations in the graph are explored with less emphasis, which offers extra robust supervision. Additionally, we optimize the relationships of neighbors at different orders through meta learning (Vanschoren, 2018) to ensure robustness.

In detail, for each unlabeled sample $\boldsymbol{x}_i^{(1)}$, we identify its $k$ nearest neighbors in $\mathcal{D}^{(2)}$, denoted as $\mathcal{N}(\boldsymbol{x}_i^{(1)})$. Similarly, we record the cross-modal neighbors of $\boldsymbol{x}_j^{(2)}$ as $\mathcal{N}(\boldsymbol{x}_j^{(2)})$. To ensure accurate relations, we construct a geometric graph to connect these unlabeled samples using mutual nearest neighbors. In other words, the adjacency matrix can be written as:

$$A_{i,j} = \begin{cases} 1 & \text{if } \boldsymbol{x}_j^{(1)} \in \mathcal{N}(\boldsymbol{x}_i^{(2)}) \wedge \boldsymbol{x}_j^{(2)} \in \mathcal{N}(\boldsymbol{x}_i^{(1)})\}, \\ 0 & \text{otherwise.} \end{cases} \tag{6}$$

With the geometric graph, we can optimize the network by maximizing the similarity of connected samples. To reduce the potential representation collapse (Chi et al., 2022), we introduce a memory bank to restore every deep representation pair as $\boldsymbol{r}_i^{(1)}$ and $\boldsymbol{r}_i^{(2)}$, which are updated using samples in the mini-batch. The optimization objective can be written as:

$$\mathcal{L}_{GEO} = -\sum_{\boldsymbol{x}_i^{(1)}\in\mathcal{B}^{(1)}}\sum_{\boldsymbol{x}_j^{(2)}\in\mathcal{B}^{(2)}} A_{ij}(\boldsymbol{z}_i^{(1)}\star\boldsymbol{r}_j^{(2)} + \boldsymbol{z}_i^{(2)}\star\boldsymbol{r}_j^{(1)}), \tag{7}$$

where $\mathcal{B}^{(1)}\subset\mathcal{D}^{(1)}$ and $\mathcal{B}^{(2)}\subset\mathcal{D}^{(2)}$ are from a mini-batch. $\star$ calculates the cosine similarity between samples. After minimizing Eqn. 7, we utilize deep features in $\mathcal{B}^{(1)}$ and $\mathcal{B}^{(2)}$ to update the memory bank. However, our geometric relations are measured a little strictly, which could filter out a range of positive pairs. To remedy this, we explore high-order geometric relations to enhance the density

of the graph. In particular, we define a second-order geometric graph with the adjacent matrix as follows:

$$A_{i,j}^S = \begin{cases} 1 & \text{if } \sum_k A_{ik} A_{kj} > 0, \\ 0 & \text{otherwise,} \end{cases} \tag{8}$$

where two samples are connected if they are both related to at least one intermediate sample. Similarly, the loss objective for learning from the second-order graph is written as:

$$\mathcal{L}_{SGEO} = - \sum_{\boldsymbol{x}_i^{(1)} \in \mathcal{B}^{(1)}} \sum_{\boldsymbol{x}_j^{(2)} \in \mathcal{B}^{(2)}} A_{ij}^S (\boldsymbol{z}_i^{(1)} \star \boldsymbol{r}_j^{(2)} + \boldsymbol{z}_i^{(2)} \star \boldsymbol{r}_j^{(1)}). \tag{9}$$

However, due to the graph expansion, there could be a few false positives introduced. To promise a robust optimization process, we expect the gradient alignment of two objectives, i.e., $\nabla \mathcal{L}_{GEO}$ and $\nabla \mathcal{L}_{SGEO}$. In particular, if we have $\nabla \mathcal{L}_{GEO}(\phi) \cdot \nabla \mathcal{L}_{SGEO}(\phi) > 0$ with network parameters $\phi$, the loss of two objective can be minimized simultaneously while $\nabla \mathcal{L}_{GEO}(\phi) \cdot \nabla \mathcal{L}_{SGEO}(\phi) \leq 0$ would fail it. Therefore, we propose a meta-learning (Finn et al., 2017; Franceschi et al., 2018) paradigm with bi-level optimization (Sinha et al., 2017). Here, minimizing Eqn. 7 and Eqn. 9 would be considered as meta-train and meta-test tasks, respectively. In the inner task, we conduct one-step gradient descent as:

$$\phi' = \phi - \alpha \nabla \mathcal{L}_{GEO}(\phi), \tag{10}$$

where $\alpha$ denotes the learning rate. In the outer task, we make a final update using the following equation:

$$\min_{\phi} \mathcal{L}_{GRM} = \mathcal{L}_{GEO}(\phi) + \lambda \mathcal{L}_{SGEO}(\phi'), \tag{11}$$

where $\lambda < 1$ is a parameter to give the priority of $\mathcal{L}_{GEO}$. With our meta-learning paradigm, we can achieve a robust interface of two different objectives, which results in a reduced semantic gap between the two modalities.

**Theoetical Analysis: A SGD Perspective.** Next, we provide the theoretical analysis of the proposed geometric relation exploration. The proof of Theorem 3.1 can be found in Appendix A.

**Theorem 3.1.** *Let $F_1(x)$ and $F_2(x)$ be two real-valued function on $\mathbb{R}^d$. Suppose $g_{1,k}(x)$ and $g_{2,k}(x)$ are two estimates of gradients at the $k$-th iteration and define $\mathcal{F}_k = \sigma(\{x_k, g_{1,k-1}, g_{2,k-1}, x_{k-1}, ..., g_{1,1}, g_{2,1}, x_0\})$. Assume there exists constants $L, M, a, \sigma^2, \zeta^2 \geq 0$ and $0 \leq m < 1$ such that*

*(1) $F(x) = F_1(x) + F_2(x)$ is $L$-smooth and $F_1(x), F_2(x)$ are both twice continuously differentiable;*

*(2) $\mathbb{E}\{g_{1,k}(x)|\mathcal{F}_k\} = \nabla F_1(x);\ \mathbb{E}\{g_{2,k}(x)|\mathcal{F}_k\} = \nabla F_2(x) + b(x)$, where $b(x) : \mathbb{R}^d \to \mathbb{R}^d$ is a bias function;*

*(3) $\mathbb{E}\{\|g_{1,k}(x) + g_{2,k}(x)\|^2|\mathcal{F}_k\} \leq M\|\nabla F(x) + b(x)\|^2 + \sigma^2$ for all $x \in \mathbb{R}^d$ for all $x \in \mathbb{R}^d$;*

*(4) $\|b(x)\|^2 \leq m\|\nabla F_1(x) + \nabla F_2(x)\|^2 + \zeta^2$ for all $x \in \mathbb{R}^d$;*

*(5) $\|\nabla^2 F_1(x)\| \leq a$ and $\|\nabla g_{2,k}(x)\|_2 \leq a$ a.s. for all $x \in \mathbb{R}^d$.*

*(6) $\langle \nabla F(x), (\nabla^2 F_2(y) + \nabla b(y)) \cdot F_1(x) \rangle \leq 0$ for all $x \in \mathbb{R}^d$ and $y$ lies between $x$ and $x - \alpha \nabla F_1(x_k)$.*

*Consider SGD updates*

$$x_{k+1} = x_k - \gamma [g_{1,k}(x_k) + g_{2,k}(x_k - \alpha \nabla F_1(x_k))]. \tag{12}$$

*Then, we have*

$$\frac{1-m}{2} \cdot \frac{1}{K+1} \sum_{k=0}^{K} \|\nabla F(x_k)\|^2 \leq 2\sqrt{\frac{\Delta_0 L(\sigma^2 + \alpha^2 a^4)}{K+1}} + \frac{2\Delta_0 LM}{K+1} + \frac{\zeta^2}{2}. \tag{13}$$

**Remarks.** Condition (1)-(4) are standard conditions for stochastic gradient descent (Ajalloeian & Stich, 2020). Condition (6) implies that the gradient of $F_1$ helps find the largely correct descent direction. For example, when $d = 1$, $F_2(x)$ is convex and $b(x)$ is decreasing, condition (6) becomes $F'(x)F_1'(x)\left(F_2''(y) + b'(y)\right) \leq 0$, which is equivalent to $F'(x)F_1'(x) \geq 0$. In our setting, this condition means that $\nabla\mathcal{L}_{GEO}$ defines the largely correct descent direction, which is consistent with our design intuition. Based on Theorem 3.1, we can have the following corollaries.

**Corollary 3.2.** *Consider our optimization problem in Eqn. 11 with $\phi' = \phi - \alpha\nabla\mathbb{E}\mathcal{L}_{GEO}(\phi)$ and assume the same conditions as Theorem 3.1 with $F_1(\phi) = \mathbb{E}\mathcal{L}_{GEO}(\phi)$ and $F_2(\phi) = \lambda\mathbb{E}\mathcal{L}_{SGEO}(\phi)$. Then, in our algorithm, SGD converges with*

$$\frac{1}{K+1}\sum_{k=0}^{K}\|\nabla F(\phi_k)\|^2 \leq O\left(\sqrt{\frac{1}{K}}\right) + O\left(\frac{1}{K}\right) + \frac{\zeta^2}{2}, \tag{14}$$

*where $F(\phi) = \mathbb{E}\mathcal{L}_{GEO}(\phi) + \lambda\mathbb{E}\mathcal{L}_{SGEO}(\phi)$.*

**Corollary 3.3.** *Suppose in Theorem 3.1, $g_{1,k}(x)$ and $g_{2,k}(x)$ have the form*

$$g_{1,k}(x) = \frac{1}{B_1}\sum_{b=1}^{B_1}\tilde{g}_{1,k}(x;\xi_{1,k}^{(b)}), \ g_{2,k}(x) = \frac{1}{B_2}\sum_{b=1}^{B_2}\tilde{g}_{2,k}(x;\xi_{2,k}^{(b)}), \tag{15}$$

*where $\xi_{1,k}^{(b)}$ and $\xi_{2,k}^{(b)}$ are independent samples and $\tilde{g}_{i,k}(x)$ satisfies condition (2) with $\mathrm{Var}\left\{\tilde{g}_{i,k}(x)|\mathcal{F}_k\right\} \leq \tilde{\sigma}_i^2$ ($i = 1, 2$). Then,*

$$\frac{1-m}{2}\cdot\frac{1}{K+1}\sum_{k=0}^{K}\|\nabla F(x_k)\|^2 \leq 2\sqrt{\frac{\Delta_0 L\left(\tilde{\sigma}_1^2/B_1 + \tilde{\sigma}_2^2/B_2 + \alpha^2 a^4\right)}{K+1}} + \frac{2\Delta_0 LM}{K+1} + \frac{\zeta^2}{2}. \tag{16}$$

*Note that $B_2 = O(n^2)$, where $n$ is the number of nodes in each mini-batch, so the inclusion of the second-order neighbors helps efficiently find the solution of optimization problem in Eqn. 11.*

## 3.4 NONPARAMETRIC SEMI-SUPERVISED LEARNING WITH CROSS-MODAL SUPERVISION

Another challenge is label scarcity, which hinders discriminative cell representations. Previous semi-supervised approaches usually generate label distributions using classifiers for pseudo-labeling (Berthelot et al., 2019a;b; Hu et al., 2021b). However, these approaches are not applicable to our representation learning framework. To tackle this, we introduce a nonparametric way to generate label distributions by comparing unlabeled cell representations with support representations, which guide the supervision across different modalities.

In particular, we sample a subset from the labeled dataset $\mathcal{S}^{(1)} \subset \mathcal{D}^{(1),l}$ as support samples and reconstruct the label distributions using the following nonparametric classifier:

$$\chi(\boldsymbol{x}_i^{(1)}) = \sum_{(\boldsymbol{x},\boldsymbol{y})\in\mathcal{S}^{(1)}}\left(\frac{\left(\boldsymbol{z}_i^{(1)}\star\phi_e^{(1)}(\boldsymbol{x})/\tau\right)}{\sum_{(\boldsymbol{x}',\boldsymbol{y}')\in\mathcal{S}^{(1)}}\boldsymbol{z}_i^{(1)}\star\phi_e^{(1)}(\boldsymbol{x}')/\tau}\right)\boldsymbol{y}, \tag{17}$$

where $\tau$ is a temperature parameter set to $0.5$ empirically as (Chen et al., 2020). Similarly, we can reconstruct the label vector for each unlabeled sample $\boldsymbol{x}_j^{(2)}$ using:

$$\chi(\boldsymbol{x}_i^{(2)}) = \sum_{(\boldsymbol{x},\boldsymbol{y})\in\mathcal{S}^{(2)}}\left(\frac{\left(\boldsymbol{z}_i^{(2)}\star\phi_e^{(2)}(\boldsymbol{x})/\tau\right)}{\sum_{(\boldsymbol{x}',\boldsymbol{y}')\in\mathcal{S}^{(2)}}\boldsymbol{z}_i^{(2)}\star\phi_e^{(2)}(\boldsymbol{x}')/\tau}\right)\boldsymbol{y}, \tag{18}$$

where $\mathcal{S}^{(2)}$ denotes the a batch of labeled samples from $\mathcal{D}^{(2),l}$. These reconstructed label vectors are likely to have high entropy when involving extensive samples with different semantics. To tackle this, we introduce a sharpening operator $\Psi(\cdot)$ for refinement. Given a label distribution $\boldsymbol{p} \in [0,1]^C$, we have

$$[\Psi(\boldsymbol{p})]_c := \frac{[\boldsymbol{p}]_c^2}{\sum_{k=1}^{K}[\boldsymbol{p}]_c^2}, c = 1,\dots,C, \tag{19}$$

where $[\boldsymbol{p}]_c$ return the $c$-th element of the vector. Our sharpening operator can increase the purification of label distributions to generate informative signals for effective supervision (Li et al., 2020). Finally, we utilize the sharpened label predictions from one modality to supervise the optimization of the other modality. In formulation, we have:

$$\mathcal{L}_{NSL} = \sum_{i=1}^{N} [H\left(\Psi\left(\boldsymbol{p}_i^{(1)}\right), \boldsymbol{p}_j^{(2)}\right) + H\left(\Psi\left(\boldsymbol{p}_i^{(2)}\right), \boldsymbol{p}_j^{(1)}\right)], \tag{20}$$

where $H(\cdot, \cdot)$ calculates the cross-entropy between two distributions. In our module, the sharpened label vectors of one modality serve as the supervision to produce semantics information for the other modality. In this manner, the label scarcity problem is overcome to some extent. Moreover, the proposed nonparametric cross-modal supervision can make use of extra information from different modalities, thus reducing potential overfitting with regularization (Chen et al., 2021).

### 3.5 SUMMERIZATION

**Consistency Learning.** Finally, we adopt cross-modal consistency learning (Feng et al., 2023; Radford et al., 2021) to enhance the discriminability of cell representations. In particular, we enforce the consistency of representation for each unlabeled pair compared with the other samples in a mini-batch. Given a mini-batch with $\mathcal{B}^{(1)}$ and $\mathcal{B}^{(2)}$, the consistency learning objective can be written as:

$$\mathcal{L}_{CL} = -\sum_{i=1}^{B} -\log \frac{exp(\boldsymbol{z}_i^{(1)} \star \boldsymbol{z}_i^{(2)}/\tau)}{\sum_{j=1}^{B} exp(\boldsymbol{z}_j^{(1)} \star \boldsymbol{z}_i^{(2)}/\tau)}, \tag{21}$$

where $B$ denotes the size of $\mathcal{B}^{(1)}$. Our consistency learning objective can maximize the mutual information between representations from two modalities (Chen et al., 2020; Zhang et al., 2021; Qin et al., 2022; Feng et al., 2023).

In summary, the final objective can be written as:

$$\mathcal{L} = \mathcal{L}_S + \mathcal{L}_{GRM} + \mathcal{L}_{NSL} + \mathcal{L}_{CL}. \tag{22}$$

During optimization, we first warm up the auto-encoder using the reconstruction loss and then conduct geometric relation exploration and nonparametric semi-supervised learning gradually. The whole algorithm of the proposed GRACE is summarized in Appendix C.

## 4 EXPERIMENT

### 4.1 EXPERIMENTAL SETTINGS

We evaluate the performance of GRACE with many state-of-the-art (SOTA) multimodal matching methods from diverse domains, including vision, text, and biology. We employ the widely recognized mean average precision (MAP) as the evaluation metric. The experiments are conducted on three public multi-omics datasets, including CITE-ASAP (Mimitou et al., 2021), snRNA-snATAC (Yao et al., 2020), and snRNA-snmC (Yao et al., 2020). More details about the implementation, datasets, and baselines can be found in Appendix D, E, and F.

### 4.2 MAIN RESULTS

**Quantitative Comparison.** Table 1 presents the results of quantitative experiments with varying numbers of labeled samples. From these results, several observations can be drawn: **Firstly**, the approaches based on image-text and 2D-3D matching outperform the previous scRNA-scATAC matching methods significantly. This is because these scRNA-scATAC matching methods solely rely on transfer learning for multimodal integration, overlooking the shared representations of different modalities in the embedding space. Using a shared encoder in scJoint and scBridge for two modalities instead of separate encoders is not conducive to learning discriminative representations. **Secondly**, all previous methods merely focus on the issue of modality matching while neglecting a more realistic problem of label scarcity. They fail to consider how to leverage unlabeled data efficiently to enhance matching performance. In contrast, we address this issue and design effective modules to solve this

Table 1: Performance evaluation (%) on three datasets with different numbers of labeled samples.

| Task | Datasets | CITE-ASAP | | | | snRNA-snATAC | | | | snRNA-snmC | | | | Avg |
|---|---|---|---|---|---|---|---|---|---|---|---|---|---|---|
| | Labels | 50 | 100 | 150 | 200 | 50 | 100 | 150 | 200 | 50 | 100 | 150 | 200 | |
| R2A | MRL (Hu et al., 2021a) | 54.03 | 63.41 | 69.02 | 70.78 | 55.88 | 70.76 | 77.34 | 80.74 | 55.35 | 65.62 | 80.15 | 84.56 | 68.97 |
| | DSCMR (Zhen et al., 2019) | 32.19 | 49.96 | 60.29 | 63.73 | 46.95 | 69.17 | 76.93 | 80.80 | 60.97 | 77.54 | 88.52 | 89.54 | 66.38 |
| | ALGCN (Qian et al., 2021) | 48.87 | 58.92 | 67.14 | 69.58 | 50.45 | 72.20 | 78.14 | 81.40 | 68.74 | 80.59 | 89.45 | 90.59 | 71.34 |
| | DA-P-GNN (Qian et al., 2022) | 39.13 | 54.40 | 65.38 | 67.69 | 50.49 | 73.51 | 80.28 | 83.48 | 67.71 | 83.89 | 90.05 | 92.54 | 70.71 |
| | DA-I-GNN (Qian et al., 2022) | 36.84 | 54.48 | 66.06 | 69.52 | 50.25 | 73.32 | 78.95 | 81.66 | 66.30 | 83.48 | 90.01 | 92.12 | 70.25 |
| | CLF (Jing et al., 2021) | 51.20 | 59.35 | 66.58 | 69.01 | 70.56 | 81.66 | 83.92 | 86.09 | 79.45 | 90.07 | 92.65 | 93.78 | 77.03 |
| | RONO (Feng et al., 2023) | 49.32 | 57.46 | 64.76 | 68.39 | 66.54 | 77.60 | 82.40 | 84.37 | 55.44 | 61.13 | 79.37 | 83.54 | 69.19 |
| | scJoint (Lin et al., 2022) | 25.00 | 26.68 | 29.50 | 33.60 | 22.71 | 28.90 | 46.17 | 53.05 | 17.32 | 32.13 | 43.40 | 42.87 | 33.44 |
| | scBridge (Li et al., 2023) | 45.99 | 48.73 | 49.17 | 51.66 | 42.88 | 46.91 | 55.86 | 64.60 | 33.85 | 39.79 | 47.63 | 61.99 | 49.09 |
| | GRACE | **65.91** | **69.10** | **74.03** | **75.12** | **78.51** | **83.16** | **85.01** | **89.10** | **84.17** | **92.10** | **94.21** | **96.02** | **82.20** |
| A2R | MRL (Hu et al., 2021a) | 51.26 | 61.63 | 68.99 | 70.51 | 57.75 | 69.42 | 73.47 | 76.08 | 50.51 | 60.79 | 76.06 | 81.91 | 66.53 |
| | DSCMR (Zhen et al., 2019) | 32.54 | 46.54 | 55.29 | 57.84 | 41.88 | 56.31 | 64.26 | 63.83 | 48.71 | 69.02 | 76.57 | 86.69 | 58.29 |
| | ALGCN (Qian et al., 2021) | 42.10 | 57.37 | 65.16 | 69.71 | 47.36 | 62.10 | 70.31 | 73.34 | 61.03 | 75.80 | 84.30 | 92.04 | 66.72 |
| | DA-P-GNN (Qian et al., 2022) | 44.58 | 55.19 | 65.83 | 68.53 | 50.02 | 72.76 | 78.75 | 82.61 | 67.70 | 83.15 | 90.42 | 92.00 | 70.96 |
| | DA-I-GNN (Qian et al., 2022) | 43.81 | 56.19 | 66.49 | 67.28 | 49.55 | 72.41 | 78.75 | 81.86 | 67.13 | 81.82 | 89.74 | 91.57 | 70.55 |
| | CLF (Jing et al., 2021) | 45.21 | 57.66 | 65.21 | 68.32 | 56.02 | 69.23 | 76.38 | 75.01 | 69.55 | 80.90 | 87.14 | 90.38 | 70.08 |
| | RONO (Feng et al., 2023) | 39.92 | 53.82 | 63.46 | 66.95 | 45.59 | 55.09 | 65.75 | 70.46 | 26.17 | 51.55 | 67.06 | 72.78 | 56.55 |
| | scJoint (Lin et al., 2022) | 28.80 | 33.57 | 40.30 | 45.01 | 15.06 | 17.80 | 46.92 | 49.42 | 25.85 | 30.70 | 47.25 | 50.20 | 35.91 |
| | scBridge (Li et al., 2023) | 51.22 | 53.14 | 56.73 | 59.50 | 37.55 | 45.74 | 52.95 | 60.18 | 34.53 | 38.71 | 49.94 | 63.78 | 50.33 |
| | GRACE | **65.46** | **68.13** | **71.99** | **75.05** | **77.46** | **82.39** | **84.03** | **86.64** | **80.67** | **87.67** | **92.29** | **94.80** | **80.55** |

problem. This is why we consistently and significantly surpass the compared approaches across different settings on the three datasets, especially when the labels are extremely scarce. **Moreover**, it is worth mentioning that some methods (e.g., ALGCN, DA-P-GNN, DA-I-GNN) introduce additional parameters such as Graph Neural Networks (GNNs) to boost performance, while we easily outperform them by a large margin with no extra parameters. This is attributed to the efficacy of nonparametric cross-modal supervision, demonstrating that our GRACE is concise yet impactful.

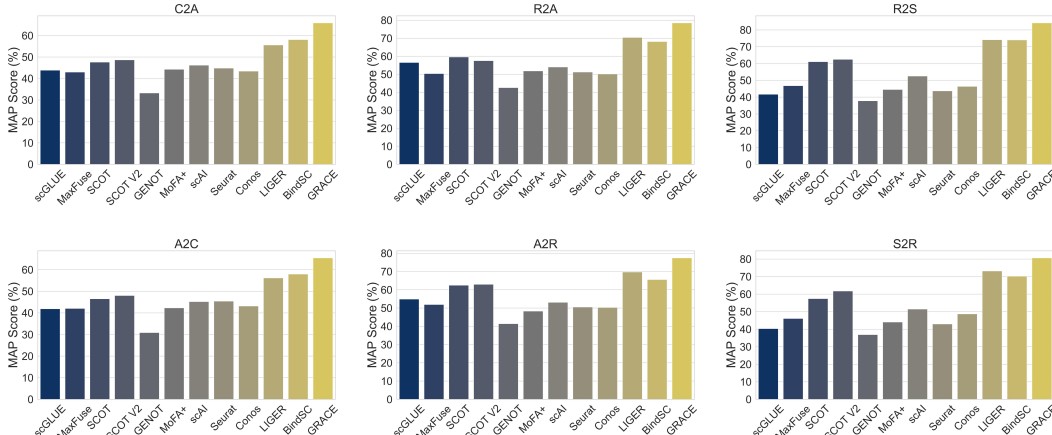

Figure 2: Further comparison with domain-specific approaches using 50 labeled samples.

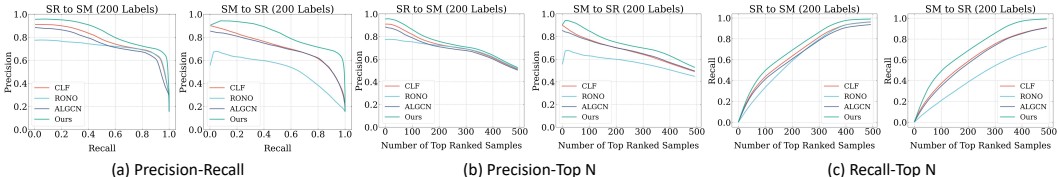

Figure 3: Three types of curves on snRNA-snmC with 200 labeled samples.

**Qualitative Comparison.** To benchmark our approach against more methods in the biological domain, we include more approaches (scGLUE (Cao & Gao, 2022), MaxFuse (Chen et al., 2024), SCOT (Demetci et al., 2022b), SCOT V2 (Demetci et al., 2022a), GENOT (Klein et al., 2023), MOFA+ (Argelaguet et al., 2020), scAI (Jin et al., 2020), Seurat (Stuart et al., 2019), Conos (Barkas et al., 2019), LIGER (Welch et al., 2019), BindSC (Dou et al., 2022)) for comparison. From the results in Figure 2, GRACE consistently outperforms the compared methods. In addition, we conduct a qualitative analysis of different approaches by plotting three types of curves in Figure 3. More

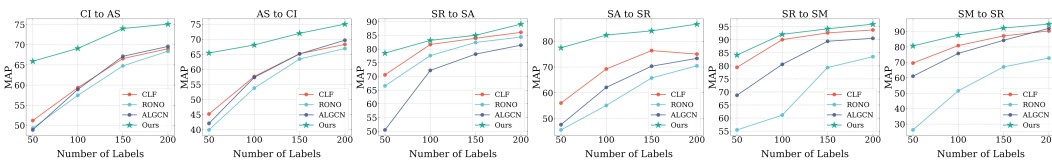

Figure 4: MAP scores with respect to the number of labeled samples on three datasets.

comprehensive qualitative results can be seen in Appendix M. The Precision-Recall curve represents the relationship between the conflicting metrics of precision and recall. The Precision-Top N and Recall-Top N curves depict the trend of precision and recall values as the top N results vary from 1 to 500 with a step size of 10. In brief, for these three types of curves, the higher-performing method's curve is positioned above the curves of other methods. It is evident that both for scRNA → scATAC and scATAC → scRNA tasks, the curve of GRACE consistently remains a significant lead over the curves of the compared baselines. Furthermore, in Figure 4, we showcase the relationship between MAP scores and the number of labeled samples. It can be observed that as the number of labeled samples increases, the performance of most of the approaches improves. Still, our GRACE outperforms all other methods significantly. Particularly in scenarios with an extremely limited number of labeled samples, such as 25, GRACE distinctly surpasses all the compared baselines.

## 4.3 DISCUSSION

**Ablation Study.** In this section, we validate the functionalities of each proposed module in Table 2. Firstly, **GRACE w/o ZN** indicates the removal of ZINB distribution to reconstruct single-cell data for warming up the auto-encoders. The results exhibit a performance decline compared to the full model.

Table 2: Ablation study (%) with 50 labeled samples.

| Model Variants | scRNA → scATAC | | | scATAC → scRNA | | |
|---|---|---|---|---|---|---|
| | CI-AS | SR-SA | SR-SM | AS-CI | SA-SR | SM-SR |
| GRACE | **65.91** | **78.51** | **84.17** | **65.46** | **77.46** | **80.67** |
| GRACE w/o ZN | 64.13 | 77.05 | 81.88 | 63.52 | 75.20 | 78.74 |
| GRACE w/o GR | 61.69 | 74.53 | 79.96 | 60.12 | 72.54 | 76.44 |
| GRACE w/o SN | 62.03 | 74.88 | 80.10 | 62.02 | 73.98 | 77.50 |
| GRACE w/o NS | 62.10 | 74.99 | 80.05 | 61.67 | 74.65 | 77.71 |

This is because the network cannot extract semantic information from the pretext task of reconstructing the original sequences. The model can only obtain parameters through random initialization, but the intrinsic distribution from random initialization often does not meet the underlying distribution of single-cell data. Furthermore, **GRACE w/o GR** removes the exploration of the geometric relation. While training with labeled data, the model has already developed some initial capacity to explore latent representations. By constructing graph-based structural relationships of high-order neighbors in the embedding space, the model further aligns the intra-class and inter-modality representations. From the results, it can be observed that this module has a significant impact on the final performance. In addition, **GRACE w/o SN** merely removes the second-order neighbors, thus the performance of this model is slightly better than the performance of the **GRACE w/o GR**. Lastly, **GRACE w/o NS** eliminates the nonparametric cross-modal supervision. By reconstructing the label vector distribution of one modality to supervise the other modality, the model can enforce consistent pseudo-labeling on the unlabeled data of both modalities, thereby enhancing cross-modal consistency. The performance decline observed after removing this module confirms its effectiveness.

**Sensitivity Analysis.** In Figure 5, we provide the sensitivity analysis of two crucial hyper-parameters $k$ and $\lambda$. Firstly, we explore the number of nearest neighbors $k$. A larger value of $k$ may include incorrect neighbors, while a smaller value of $k$ may neglect potential correct neighbors. We gradually increase $k$ from 1 to 6 and observe that the performance is worse when $k = 1$. As $k$ increases, the performance improves and eventually saturates. Based on this analysis, we determine the optimal value of $k = 5$. Next,

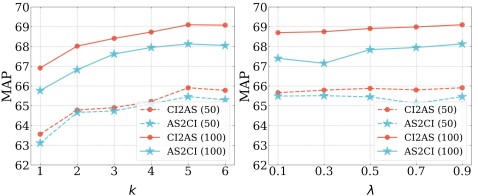

Figure 5: Sensitivity of $k$ and $\lambda$ with 50 and 100 labeled samples on CITE-ASAP.

with $k$ fixed at 5, we investigate the sensitivity of $\lambda$. $\lambda$ is less than 1, indicating that we consider the second-order neighbors less accurate than the first-order neighbors. Assigning a smaller coefficient to

the second-order neighbors prioritizes the first-order neighbors. We increase $\lambda$ from 0.1 to 0.9 and observe that the performance fluctuates within a small range. This indicates that the model is not highly sensitive to the value of $\lambda$. Therefore, we set the default value of $\lambda$ as 0.9.

**Visualization.** In Figure 6, we visualize the embeddings using t-SNE (Van der Maaten & Hinton, 2008). The scRNA-seq and scATAC-seq embeddings are colored yellow and blue, respectively. The overlap degree reflects the extent to which multimodal representations are aligned. It can be observed that the embeddings of both modalities in ALGCN are unevenly distributed and have limited overlaps, whereas GRACE achieves better alignment in the embeddings. This further validates the success of our proposed geometric relation exploration in multimodal data matching under label scarcity conditions.

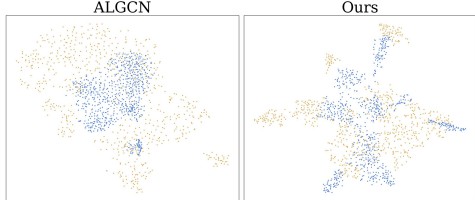

Figure 6: The t-SNE visualization with 50 labeled samples on CITE-ASAP.

### 4.4 FURTHER EXPLORATION ON BIOLOGICAL APPLICATIONS

The proposed GRACE is a versatile representation learning framework as it learns high-quality representations across multi-omics data. Therefore, it is scalable to diverse multi-omics single-cell data analysis tasks. In this section, we investigate the potential biological applications of GRACE.

**Multi-omics data integration.** In Table 3, we conduct an experiment on multi-omics data integration using CITE-ASAP data. PC and FM are short for Pearson Correlation and FOSCTMM, respectively. The results show that our method outperforms the compared baselines in terms of both evaluation metrics, indicating that GRACE successfully integrates CITE-seq and ASAP-seq data.

Table 3: Results (%) on multi-omics data integration.

| Labels | 50 | | 100 | | 150 | | 200 | |
|---|---|---|---|---|---|---|---|---|
| Metric | PC $\uparrow$ | FM $\downarrow$ | PC $\uparrow$ | FM $\downarrow$ | PC $\uparrow$ | FM $\downarrow$ | PC $\uparrow$ | FM $\downarrow$ |
| scJoint | 58.88 | 24.40 | 63.41 | 23.21 | 65.15 | 21.00 | 69.79 | 20.04 |
| MRL | 65.61 | 21.39 | 68.77 | 20.05 | 71.49 | 19.94 | 74.55 | 18.26 |
| Ours | **71.53** | **18.06** | **74.08** | **17.22** | **75.66** | **16.91** | **77.83** | **15.99** |

**Batch effect correction.** In Table 4, we explore the scalability of GRACE to batch effect correction task on multi-batch Mouse Atlas data. V1-V6 denote the index of different batches. The results demonstrate besides multimodal single-cell data, GRACE can also align scRNA-seq data from different batches to reduce batch effects. From the results, it can be found that GRACE significantly corrects batch effects, thereby achieving better performance.

Table 4: MAP (%) on batch effect correction.

| Batch | V1-V2 | V2-V1 | V3-V4 | V4-V3 | V5-V6 | V6-V5 |
|---|---|---|---|---|---|---|
| scJoint | 84.68 | 83.95 | 82.82 | 81.54 | 87.01 | 86.45 |
| MRL | 92.46 | 91.07 | 91.87 | 90.65 | 94.85 | 93.53 |
| Ours | **98.63** | **98.88** | **95.77** | **93.11** | **96.52** | **96.51** |

**Label transfer.** In Table 5, we showcase the label transfer results from CITE-seq to ASAP-seq data. It can be observed that the proposed representation learning framework GRACE exceeds the previous SOTA domain-specific methods (scJoint Lin et al. (2022), scBridge (Li et al., 2023), scNCL (Yan et al., 2023)) with different numbers of labeled scRNA-seq samples. The results indicate that even without labeled scATAC-seq samples for semantics injection, GRACE still can successfully transfer cell type knowledge information.

Table 5: Acc (%) on label transfer.

| Label Ratio | 1% | 5% | 10% |
|---|---|---|---|
| scJoint | 62.36 | 67.75 | 70.02 |
| scBridge | 64.99 | 69.86 | 71.44 |
| scNCL | 61.50 | 69.04 | 71.17 |
| Ours | **66.34** | **70.78** | **72.01** |

### 5 CONCLUSION

This paper investigates an underexplored problem of realistic multimodal single-cell data matching and proposes a novel approach named GRACE, which maps high-dimensional multimodal biological data into a common embedding space under label scarcity conditions. Extensive experiments conducted on several benchmark datasets verify the superiority of the proposed method over numerous baselines and prove that GRACE is scalable to diverse multi-omics single-cell data analysis tasks. In future works, we plan to extend our framework to a broader range of applications such as spatial transcriptomics single-cell data analysis and multimodal single-cell foundation models.

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

## A    PROOF OF THEOREM

*Proof of Theorem 3.1.* Define $y_k = x_k - \alpha \nabla F_1(x_k)$. Note that $F_1(x) + F_2(x)$ is $L$-smooth, so

$$\mathbb{E}\left\{F(x_{k+1})|\mathcal{F}_k\right\}$$

$$\leq F(x_k) + \mathbb{E}\left\{\langle \nabla F(x_k), x_{k+1} - x_k \rangle |\mathcal{F}_k\right\} + \frac{L}{2}\mathbb{E}\left\{\|x_{k+1} - x_k\|^2 |\mathcal{F}_k\right\}$$

$$\leq F(x_k) - \gamma \langle \nabla F(x_k), \nabla F_1(x_k) + \nabla F_2(y_k) + b(y_k)\rangle + \frac{L\gamma^2}{2}\mathbb{E}\left\{\|g_{1,k}(x_k) + g_{2,k}(y_k)\|^2 |\mathcal{F}_k\right\}$$

$$\leq F(x_k) - \gamma \langle \nabla F(x_k), \nabla F(x_k) + b(x_k) - \alpha \nabla^2 F_2(\eta_k) \cdot \nabla F_1(x_k) - \alpha \nabla b(\eta_k) \cdot \nabla F_1(x_k)\rangle$$

$$+ L\gamma^2 \left\{M \|\nabla F_1(x) + \nabla F_2(x) + b(x)\|^2 + \sigma^2 + \alpha^2 a^4\right\}, \tag{23}$$

where $\eta_k$ lies between $x_k$ and $y_k$.

Therefore, using condition (6) when $\gamma < \frac{1}{2LM}$, we have

$$\mathbb{E}\left\{F(x_{k+1})|\mathcal{F}_k\right\}$$

$$\leq F(x_k) + \frac{\gamma}{2}\left\{-2\langle \nabla F(x_k), \nabla F(x_k) + b(x_k)\rangle + 2LM\gamma \|\nabla F(x_k) + b(x_k)\|^2\right\}$$

$$+ L\gamma^2 \left(\sigma^2 + \alpha^2 a^4\right)$$

$$\leq F(x_k) + \frac{\gamma}{2}\left\{-2\langle \nabla F(x_k), \nabla F(x_k) + b(x_k)\rangle + \|\nabla F(x_k) + b(x_k)\|^2\right\} + L\gamma^2 \left(\sigma^2 + \alpha^2 a^4\right)$$

$$= F(x_k) + \frac{\gamma}{2}\left\{-\|\nabla F(x_k)\|^2 + \|b(x_k)\|^2\right\} + L\gamma^2 \left(\sigma^2 + \alpha^2 a^4\right)$$

$$\leq F(x_k) + \frac{\gamma}{2}(m-1)\|\nabla F(x_k)\|^2 + \frac{\gamma\zeta^2}{2} + L\gamma^2 \left(\sigma^2 + \alpha^2 a^4\right) \quad \text{(condition (4))} \tag{24}$$

Hence, taking expectation on both sides, we get

$$\frac{\gamma}{2}(1-m)\|\nabla F(x_k)\|^2 \leq \left\{F(x_k) - F(x_{k+1})\right\} + \frac{\gamma\zeta^2}{2} + L\gamma^2 \left(\sigma^2 + \alpha^2 a^4\right). \tag{25}$$

Taking averaging over $k = 0, 1, \ldots, K$, we have

$$\frac{1}{K+1}\sum_{k=0}^{K}\|\nabla F(x_k)\|^2 \leq \frac{2\left\{F(x_0) - F^*\right\}}{\gamma(1-m)(K+1)} + \frac{\zeta^2 + 2L\gamma\left(\sigma^2 + \alpha^2 a^4\right)}{1-m}, \tag{26}$$

where $F^* = \min_x F(x)$. Set $\Delta_0 = F(x_0) - F^*$ and $\gamma = \left\{\left(\frac{\Delta_0}{L(K+1)(\sigma^2+\alpha^2 a^4)}\right)^{-1/2} + 2LM\right\}^{-1}$,

then we have

$$\frac{1-m}{2} \cdot \frac{1}{K+1}\sum_{k=0}^{K}\|\nabla F(x_k)\|^2 \leq 2\sqrt{\frac{\Delta_0 L\left(\sigma^2 + \alpha^2 a^4\right)}{K+1}} + \frac{2\Delta_0 LM}{K+1} + \frac{\zeta^2}{2}. \tag{27}$$

$\square$

## B    RELATED WORK

**Multimodal Single-cell Data Integration.** Integrating multimodal single-cell data from various sources is an essential problem for understanding biological processes. Early attempts usually focus on matrix factorization (Duren et al., 2018; Jin et al., 2020; Stein-O'Brien et al., 2018) and statistical models (Shen et al., 2009; Stuart et al., 2019; Welch et al., 2017). Matrix factorization provides an effective tool for generating low-dimensional features from high-dimensional omics data while statistical models usually introduce a range of assumptions about underlying distributions for probabilistic inference. For example, scMoMaT (Zhang et al., 2023b) adopts matrix tri-factorization to identify multimodal biomarkers associated with cell types. Recently, deep learning-based methods have achieved extensive progress including auto-encoder-based methods and graph neural network-based methods. scMoGNN (Wen et al., 2022a) constructs a graph model to depict the correlation

between genes and cells and then employs cell-feature graph convolution to learn discriminative cell and gene representations. Nevertheless, these approaches are data-hungry with the requirement of extensive labeled data, which is hard to get in realistic scenarios. Towards this end, this work for the first time, centers on a practical issue of semi-supervised multimodal single-cell data matching and proposes an effective approach dubbed GRACE to solve the problem.

**Semi-supervised Learning.** Due to its ability to address label scarcity, semi-supervised learning (Wu et al., 2023; Assran et al., 2021) has gained significant attention with wide applications including semantic segmentation (Qiao et al., 2023; Chen et al., 2023) and object detection (Hua et al., 2023; Zhang et al., 2023a). Research semi-supervised learning approaches can be broadly categorized as consistency regularization approaches (Miyato et al., 2018; Sohn et al., 2020; Xie et al., 2020) and pseudo-labeling approaches (Berthelot et al., 2019a;b; Hu et al., 2021b). Pseudo-labeling approaches annotate unlabeled samples using a weak model and incorporate confident pseudo-labels and their corresponding sample into the training set. Moreover, dual learning, dynamical thresholding, and curriculum learning are utilized to ensure accurate and unbiased pseudo-labels for reliable optimization. In contrast, consistency regularization approaches usually incorporate perturbation of various sources including input (Xie et al., 2020), network parameters (Ouali et al., 2020) and deep features (Ke et al., 2019), and then encourage model invariance under perturbation. Nevertheless, these approaches primarily focus on single-modality classification problems, which are not applicable to multimodal data matching. In this work, we propose a new nonparametric strategy to reconstruct informative and reliable label vectors for discriminative learning across modalities.

## C  ALGORITHM

The step-by-step training algorithm of our GRACE is summarized in Algorithm 1.

---

**Algorithm 1** Training Algorithm of GRACE

---

**Require:** The training dataset $\mathcal{D}^{(1)}$ and $\mathcal{D}^{(2)}$; Balance coefficient $\lambda$; Number of neighbors $k$.
**Ensure:** Two projectors $f_e^{(1)}(\cdot)$ and $f_e^{(2)}(\cdot)$;
 1: Warm up the network using Eqn. 4;
 2: Construct the memory bank using current cell representations;
 3: **repeat**
 4:     Update the geometric graphs using Eqn. 7 and Eqn. 9;
 5:     **for** $t = 1, 2, \cdots, T$ **do**
 6:         Sample a mini-batch from $\mathcal{D}^{(1)}$ and $\mathcal{D}^{(2)}$;
 7:         Generate cell representations by propagating the networks;
 8:         Conduct one-step gradient descent using Eqn. 10;
 9:         Calculate the final loss using Eqn. 22;
10:         Update the network parameters using backpropagation;
11:         Update the memory using the current mini-batch;
12:     **end for**
13: **until** convergence

---

## D  IMPLEMENTATION DETAILS

For fair comparisons, we re-implement all the baselines according to the settings in the corresponding papers. To extend these approaches to our task, we replace the encoders with two-layer MLPs. All the experiments are conducted in Pytorch with NVIDIA Tesla A100 GPUs. We utilize the SGD optimizer with a learning rate of $1e-3$ and a batch size of $64$. The baselines are trained for $50$ epochs, while our GRACE is trained for $20$ epochs. The embedding dimensions of both modalities are fixed at $256$. The number of the returned samples to a query is set to the size of the test set.

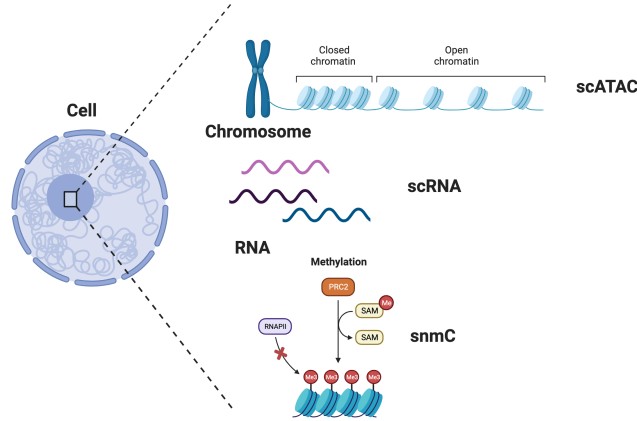

Figure 7: An illustration of various multimodal single-cell data.

# E    INTRODUCTION TO THE DATASETS

We validate the performance of GRACE on three public multi-omics datasets (Figure 7), including CITE-ASAP (Mimitou et al., 2021), snRNA-snATAC (Yao et al., 2020), and snRNA-snmC (Yao et al., 2020). We integrate the datasets into multi-omics pairs and randomly split the datasets into two parts: 80% for training and 20% for testing. Here we provide a detailed introduction to the dataset information used in this paper:

- **CITE-ASAP** (Mimitou et al., 2021) is generated from control condition. CITE-seq is a technology that enables simultaneous profiling of gene expression and protein abundance. Similarly, ASAP-seq allows for concurrent profiling of accessible chromatin and protein levels in thousands of single cells. The whole dataset contains 4,644 cells from CITE-seq and 4502 cells from ASAP-seq. The lengths of both CITE-seq and ASAP-seq are 17,441. After preprocessing, we get 3,662 CITE-ASAP pairs from 7 cell types.

- **snRNA-snATAC** (Yao et al., 2020) and **snRNA-snmC** (Yao et al., 2020) are generated from mouse primary motor cortex. The lengths of snRNA-seq, snATAC-seq, and snmC-seq data are 18,603. The entire dataset consists of 16,624 snRNA-seq, 7,962 snATAC-seq, and 9,633 snmC-seq. After manually excluding samples with different cell types, we assemble a dataset consisting of 7,904 pairs of snRNA-snATAC samples from 18 different cell types, as well as 8,270 snRNA-snmC samples from 17 cell types.

# F    INTRODUCTION TO THE BASELINES

Our GRACE is compared with many SOTA multimodal matching methods, including seven methods from visual and textual domain (MRL (Hu et al., 2021a), DSCMR (Zhen et al., 2019), ALGCN (Qian et al., 2021), DA-P-GNN (Qian et al., 2022), DA-I-GNN (Qian et al., 2022), CLF (Jing et al., 2021), and RONO (Feng et al., 2023)), and two methods from biological domain (scJoint (Lin et al., 2022), scBridge (Li et al., 2023)). The introduction to the baselines is as below:

- **MRL** (Hu et al., 2021a) utilizes multimodal robust learning to map diverse modalities into a shared latent space, which is effective against label noise.

- **DSCMR** (Zhen et al., 2019) aims to jointly minimize both the discrimination loss and the modality invariance loss, enabling the learning of shared representations for diverse modalities.

- **ALGCN** (Qian et al., 2021) retains the cross-modal semantic correlations and uncovers the latent semantic structure of labels through the joint training of two branches.

- **DA-I-GNN** (Qian et al., 2022) utilizes an Iterative Graph Neural Network (GNN) and incorporates multi-label contrastive learning to acquire a shared representation for cross-modal retrieval.

- **DA-P-GNN** (Qian et al., 2022) is similar to DA-I-GNN and employs a Probabilistic GNN.

- **CLF** (Jing et al., 2021) facilitates the learning of discriminative and modality-invariant features through a cross-modal center loss.
- **RONO** (Feng et al., 2023) incorporates a robust discriminative center learning and a shared space consistency learning mechanism for mapping different modalities into a common space against label noise.
- **scJoint** (Lin et al., 2022) integrates scRNA-seq and scATAC-seq data through transfer learning and pseudo-labeling.
- **scBridge** (Li et al., 2023) mines cross-omic samples for dataset expansion and heterogeneously integrates scRNA-seq and scATAC-seq data.

## G ASSESS THE VARIABILITY OF THE PROPOSED APPROACH

We conduct multiple trial experiments with different random seeds to assess the variability of our method and the sensitivity of our method to different random initializations. The results in Table 6 validate the superiority and robustness of our approach.

Table 6: Multiple trial comparisons on different tasks with 200 labeled samples.

| Task | C2A | A2C | R2A | A2R | R2S | S2R |
|------|-----|-----|-----|-----|-----|-----|
| scJoint | 32.94±1.21 | 44.67±0.56 | 53.68±0.45 | 49.50±0.39 | 42.66±0.38 | 50.88±0.51 |
| MRL | 70.42±0.94 | 70.10±0.87 | 80.56±1.16 | 76.81±0.73 | 84.18±0.77 | 82.11±0.31 |
| Ours | **75.06±0.05** | **74.95±0.35** | **88.79±0.24** | **86.21±0.30** | **95.97±0.06** | **94.58±0.40** |

## H INSTANCE-LEVEL MATCHING RESULTS

The proposed approach is a generalized multimodal single-cell data integration framework, which is not limited to coarse matching based on cell types, but also effective in instance-level matching. From the results in Table 7, it can be observed that GRACE performs well on paired scRNA-seq and scATAC-seq data of A549 cells

Table 7: The performance comparison of instance-level matching.

| Task | R2A | R2A | R2A | A2R | A2R | A2R |
|------|-----|-----|-----|-----|-----|-----|
| Metric | Recall@1 | Recall@5 | Recall@10 | Recall@1 | Recall@5 | Recall@10 |
| scAI | 90.12 | 92.23 | 95.87 | 89.01 | 92.53 | 95.49 |
| MRL | 90.48 | 92.14 | 96.05 | 90.85 | 92.77 | 95.10 |
| Ours | **93.66** | **94.40** | **98.79** | **93.08** | **95.69** | **97.31** |

## I PERFORMANCE COMPARISON ON FULL LABELED DATA

Our method is not only effective under label scarcity but also demonstrates excellent performance when labels are abundant. As shown in Table 8, even when using fully labeled data, our method still outperforms the compared baseline methods.

Table 8: The performance comparison of instance-level matching.

| Task | R2A | R2A | R2A | A2R | A2R | A2R |
|------|-----|-----|-----|-----|-----|-----|
| scJoint | 71.97 | 70.46 | 82.40 | 81.08 | 86.11 | 85.60 |
| MRL | 80.68 | 80.01 | 93.55 | 91.99 | 97.54 | 95.06 |
| Ours | **81.90** | **80.87** | **94.33** | **93.48** | **98.70** | **96.59** |

## J   SENSITIVITY ANALYSIS OF LOSS WEIGHT

Here, we include the sensitivity analysis of the weights for each proposed loss function. The results in Figure 8 indicate that the weights of the four losses have a relatively minor impact on the performance. Therefore, for the sake of simplicity, we set the weights for all losses to 1.

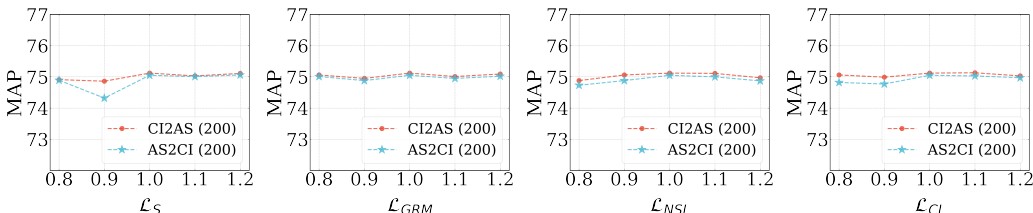

Figure 8: The sensitivity analysis of the weights for different loss functions.

## K   IMPACT OF CONSISTENCY LEARNING

Consistency learning would reduce the distance between these paired cell representations and increase the distance between unpaired cell representations, encouraging discriminative and modality-invariant representations. We have included a model variant GRACE w/o CL with 50 labeled samples to support our point. The results in Figure 9 validate the effectiveness of consistency learning.

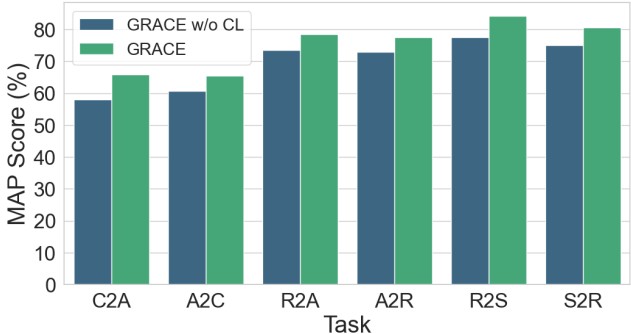

Figure 9: Ablation on the consistency learning module.

## L   COMPREHENSIVE VISUALIZATION RESULTS

Moreover, we make the comprehensive t-SNE (Van der Maaten & Hinton, 2008) visualization (Figure 10). It can be seen that compared with the other three approaches, the multimodal embeddings generated by our GRACE strike a balance between being discriminative and modality-invariant.

In addition, we also make visualizations of the feature space before and after applying the ZINB distribution reconstruction in Figure 11. From the results, we can observe the feature distribution is more discriminative after the ZINB distribution reconstruction.

Next, we provide the training curves regarding the incorporation of high-order geometric relations in Figure 12. The results show that the model achieved a MAP score of 90.17 after incorporating high-order geometric relations, surpassing the non-incorporated model which scores 88.39. This phenomenon validates that high-order geometric relations could enhance optimization robustness.

## M   COMPREHENSIVE QUALITATIVE RESULTS

We conduct comprehensive qualitative experiments, including Precision-Recall curves with different numbers of labeled samples with results shown in Figure 13, Figure 14, Figure 15, Figure 16,

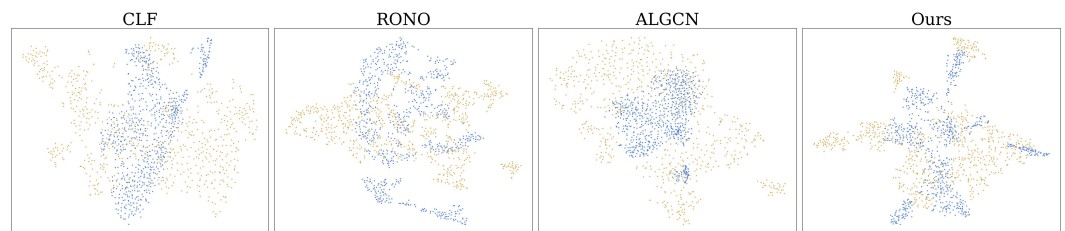

Figure 10: The t-SNE visualization of four methods. The scRNA-seq embeddings are colored yellow and the scATAC-seq embeddings are colored blue.

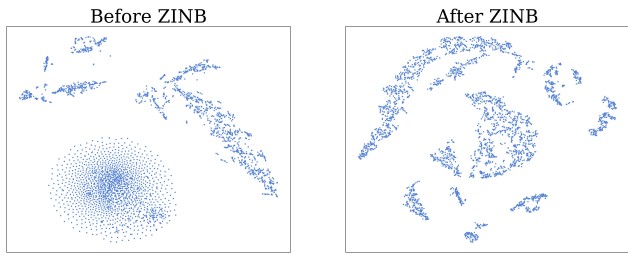

Figure 11: The t-SNE visualization of the feature space before and after the ZINB distribution reconstruction.

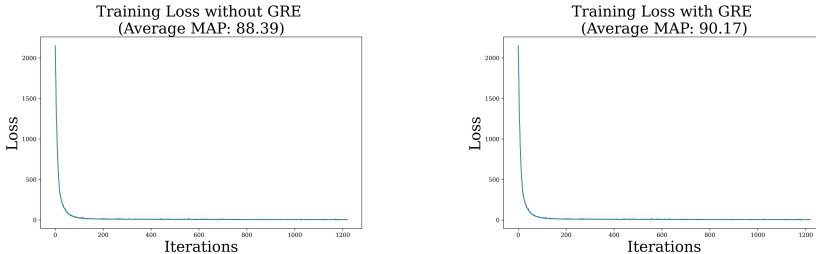

Figure 12: The training curve with respect to the high-order geometric relations.

Precision-Top N curves with different numbers of labeled samples with results shown in Figure 17, Figure 18, Figure 19, Figure 20, and Recall-Top N curves with different numbers of labeled samples with results shown in Figure 21, Figure 22, Figure 23, Figure 24. All of these qualitative experimental results consistently demonstrate that our approach is superior to the compared baselines.

## N    BROADER IMPACTS AND LIMITATIONS

In this paper, we propose a semi-supervised framework for realistic multimodal single-cell data matching, which effectively integrates multimodal biological representations. It provides inspiration for many downstream tasks in multi-omics single-cell data analysis, such as multi-omics data integration, batch effect correction, and cell type transfer. To our knowledge, no potential negative impacts or limitations have been identified.

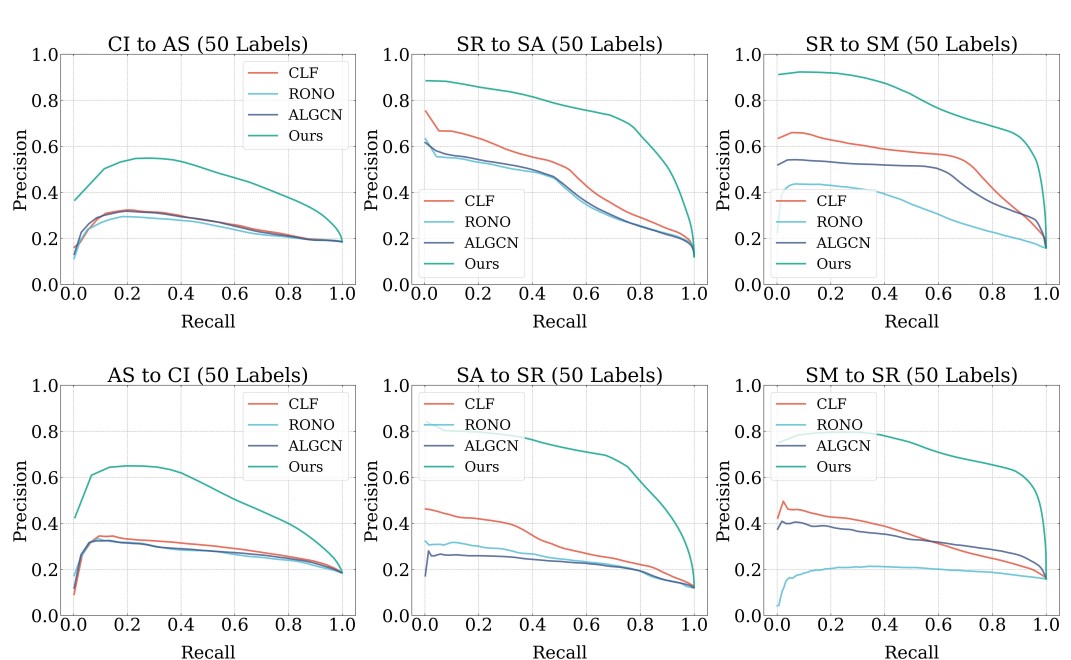

Figure 13: The Precision-Recall curves with 50 labeled samples.

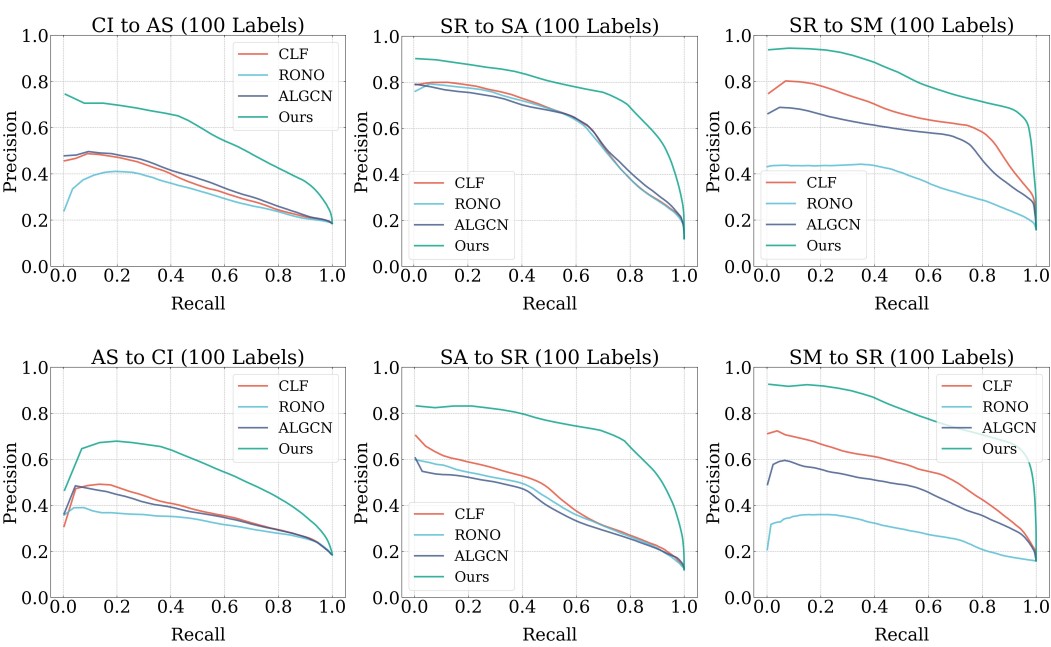

Figure 14: The Precision-Recall curves with 100 labeled samples.

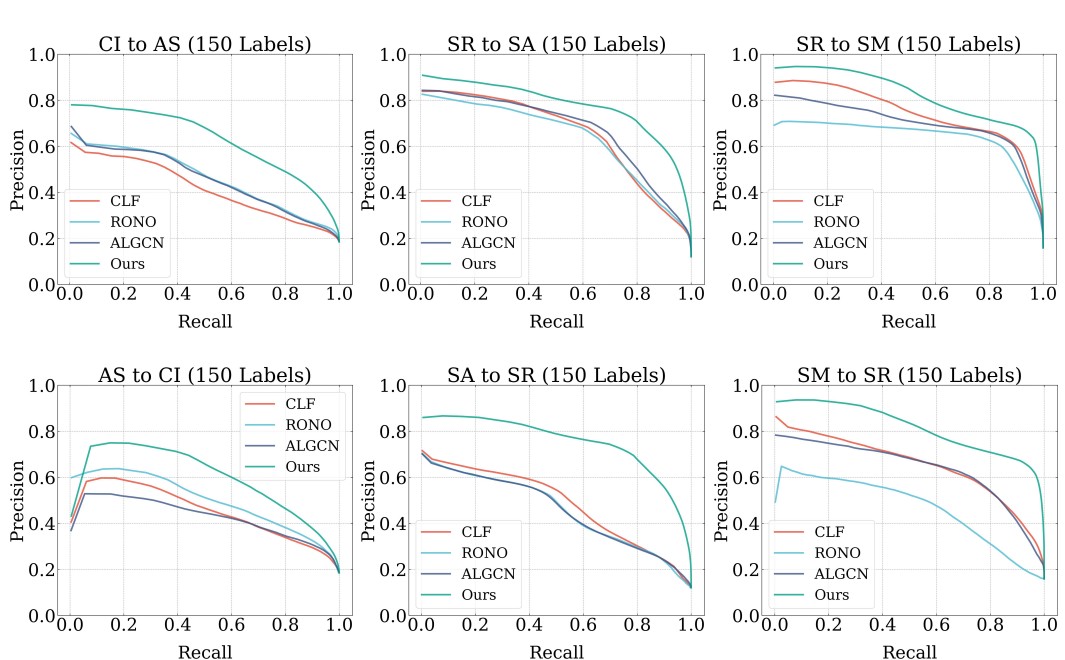

Figure 15: The Precision-Recall curves with 150 labeled samples.

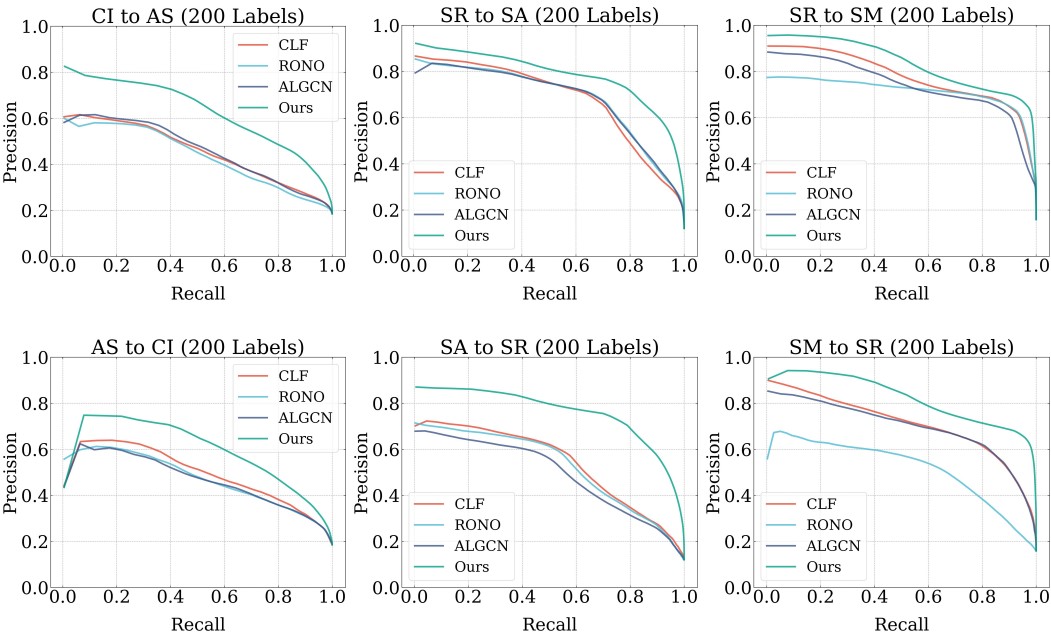

Figure 16: The Precision-Recall curves with 200 labeled samples.

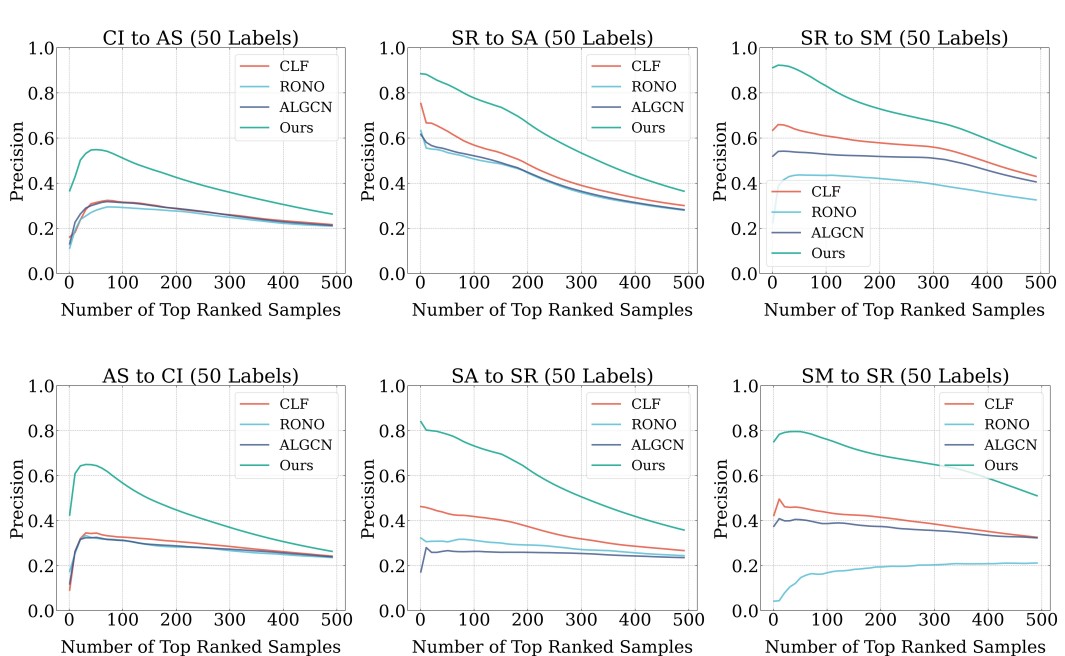

Figure 17: The Precision-Top N curves with 50 labeled samples.

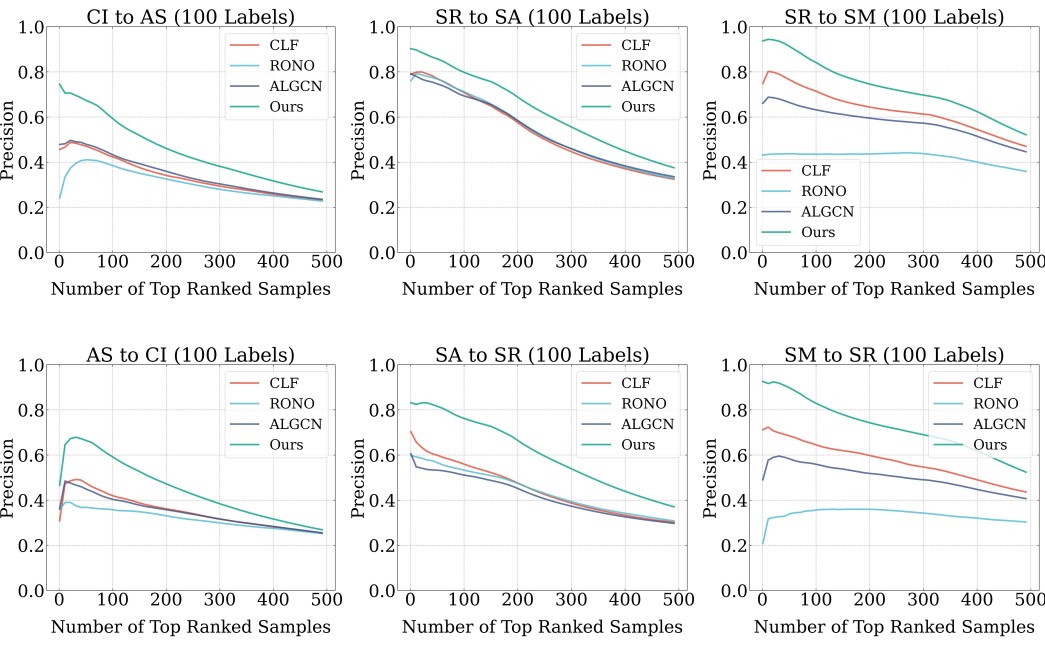

Figure 18: The Precision-Top N curves with 100 labeled samples.

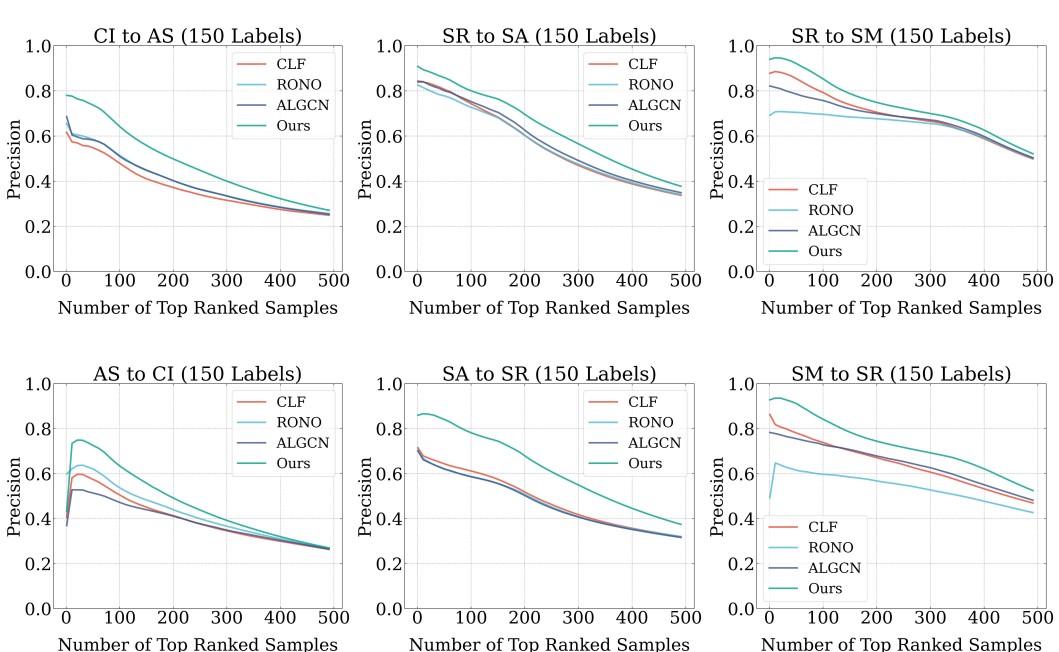

Figure 19: The Precision-Top N curves with 150 labeled samples.

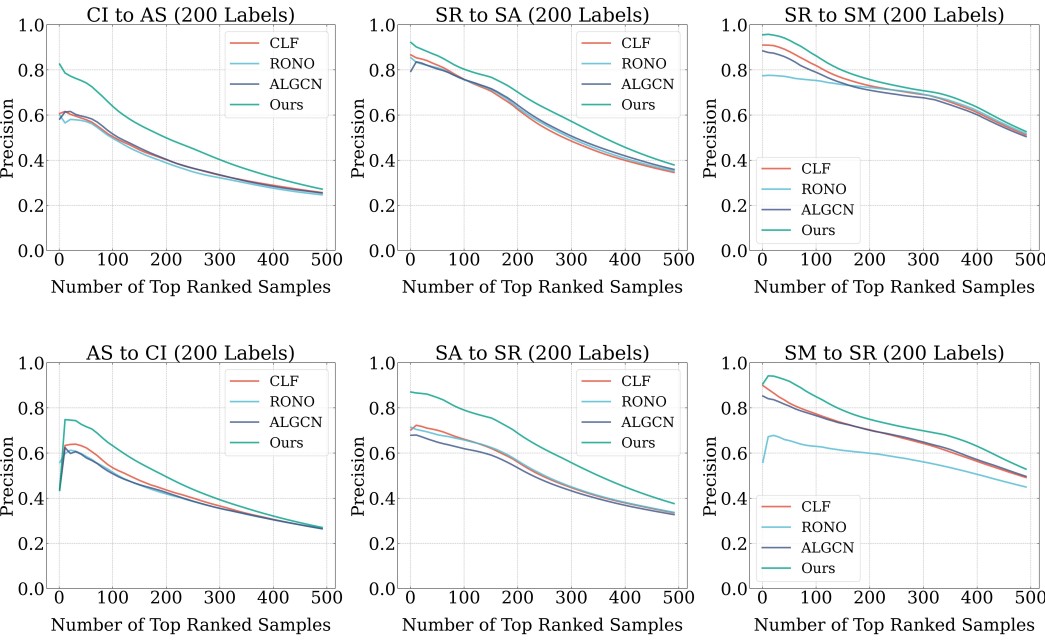

Figure 20: The Precision-Top N curves with 200 labeled samples.

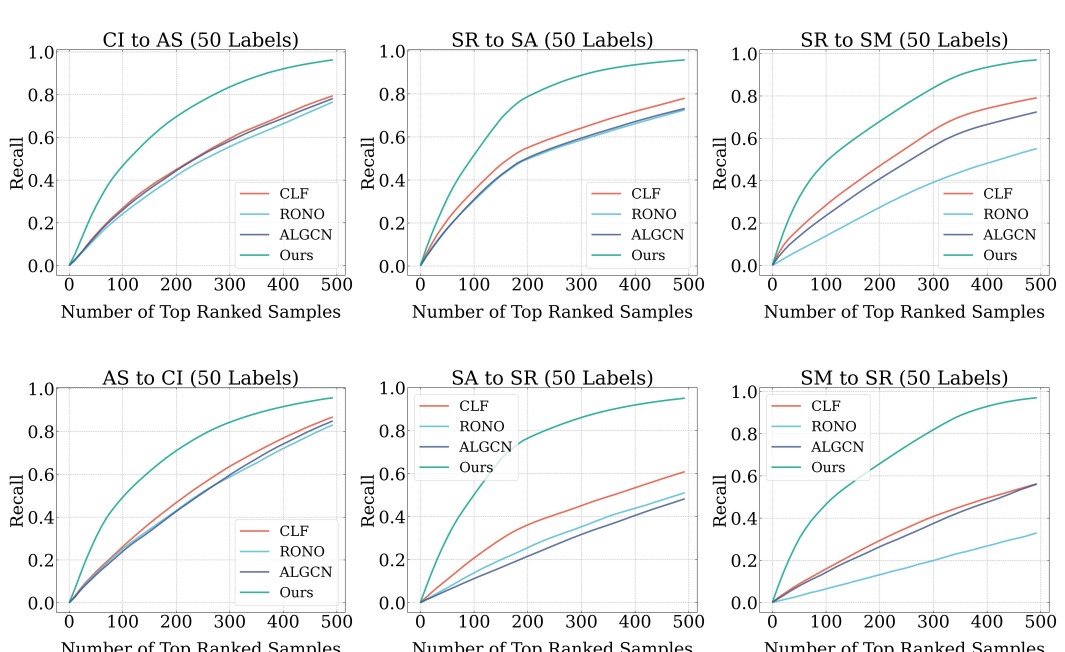

Figure 21: The Recall-Top N curves with 50 labeled samples.

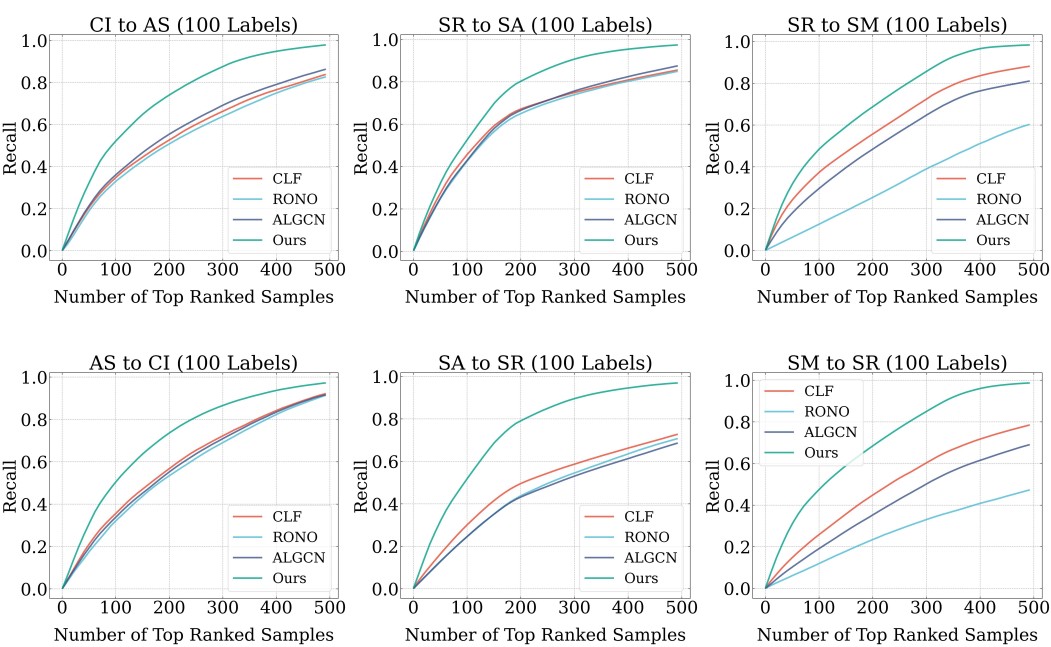

Figure 22: The Recall-Top N curves with 100 labeled samples.

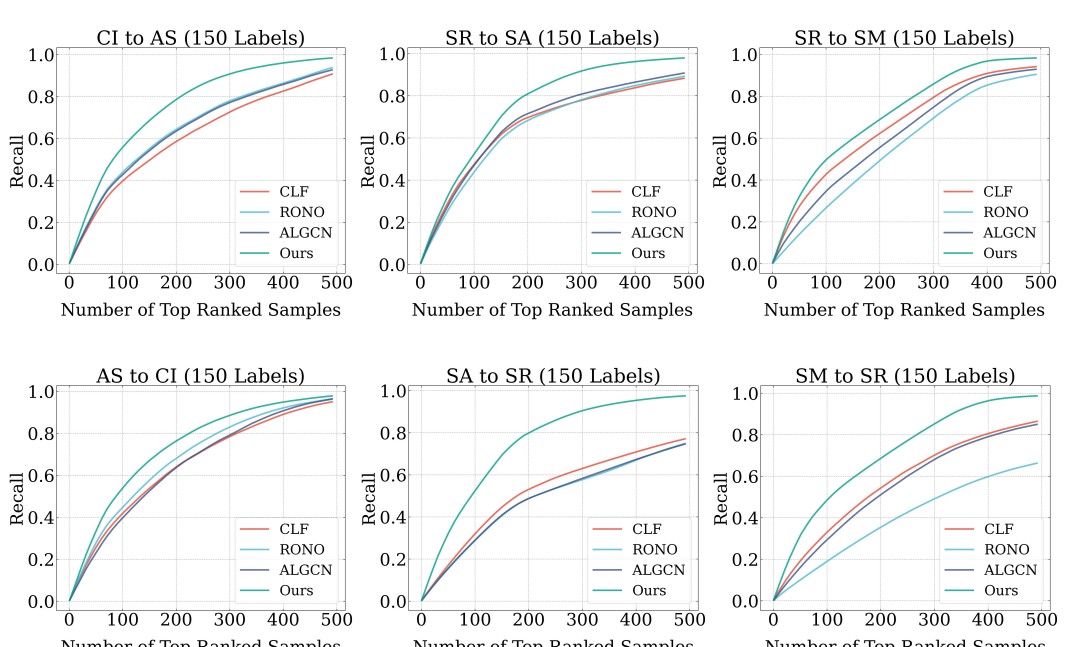

Figure 23: The Recall-Top N curves with 150 labeled samples.

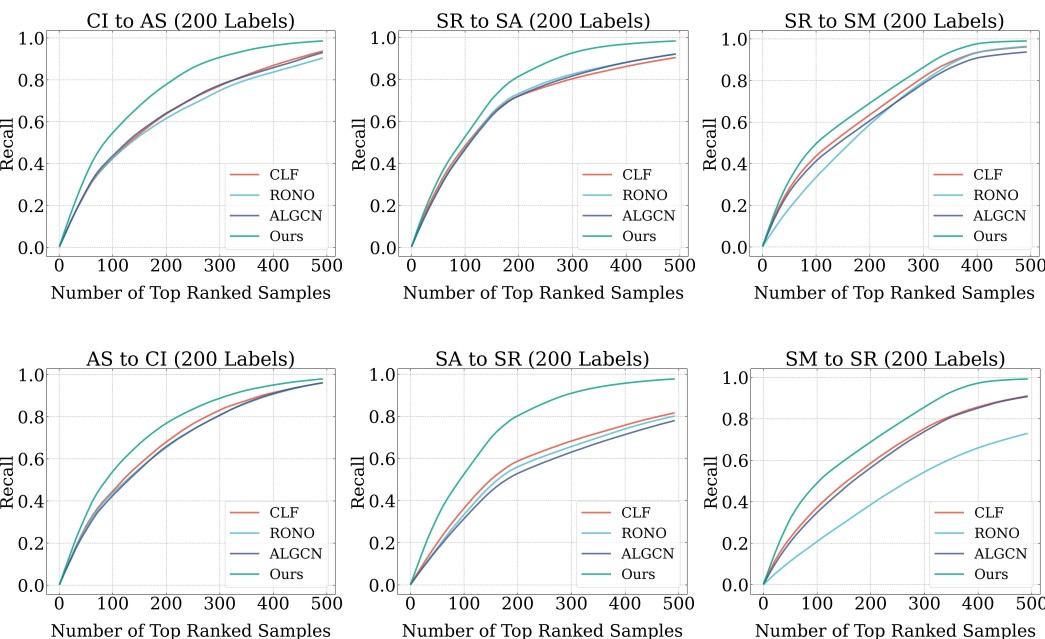

Figure 24: The Recall-Top N curves with 200 labeled samples.

