# OpenReview forum: "GRACE: Towards Realistic Multimodal Single-Cell Data Matching"
_ICLR.cc/2025/Conference — ICLR 2025 Conference Withdrawn Submission_

### Official Review · Reviewer_4QG9 · 2024-10-28

**Soundness:** 2
**Presentation:** 3
**Contribution:** 2
**Rating:** 3
**Confidence:** 2

**Summary:**

In response to the real-world scenario where labeled data is scarce, GRACE generates label vectors in a non-parametric manner, providing cross-modal supervision which contributes to the improvement of data matching performance.

**Strengths:**

The paper proposes a new method called GRACE (Geometric Relation Exploration with Cross-modal Supervision) to address the matching problem of multimodal single-cell data. The format of this paper is excellent, with clear figures and easy to read.

**Weaknesses:**

1. GRACE employs a non-parametric strategy to generate label distributions for unlabeled cell data. This is achieved by comparing the representations of unlabeled cells with support representations to provide guidance signals for cross-modal consistency. However, the quality of these support representations can significantly impact the label generation process for unlabeled samples, thereby affecting the model's final performance. If the distribution of the data used to construct the support representations is not comprehensive enough to cover all datasets, it may lead to inaccurate reconstruction of the label distributions for unlabeled samples, which in turn affects the quality of the representations learned by the model. In such cases, the model's performance may suffer as it relies on these potentially inaccurate label distributions for cross-modal consistency learning. Therefore, the paper should provide more discussion and analysis regarding this issue. Additionally, it is recommended to cite more literature related to this problem.

2. It is suggested to test performance using labeled data sampled from different subsets of cell types, or to analyze how performance varies with different sizes of labeled datasets.

3. The related work section lacks a thorough introduction to the current problem, and does not sufficiently analyze the advantages and disadvantages of the model structure presented in this paper for solving the current problem, i.e., there is a lack of necessary motivation.

4. The paper does not discuss the computational complexity of the GRACE method in detail (parameters, FLOPS), which could pose challenges for laboratories or research environments with limited resources. Please provide specific detailed information:
(1) The number of parameters in their model compared to baseline methods
(2) Training and inference times on standard hardware
(3) Memory requirements for different dataset sizes
(4) Any analysis of how computational requirements scale with dataset size or number of cell types This would help readers assess the practical applicability of the method.

5. Although sensitivity analysis of the parameters has been conducted, the selection of optimal parameters for new datasets remains an open question. Please explain whether any automatic parameter tuning methods have been explored, or if a guide can be provided for selecting parameters based on the characteristics of the dataset. Additionally, please discuss the performance stability across a broader range of parameter values.

**Questions:**

Please evaluate the reliability and robustness of the support expressions obtained from a small amount of labeled data, and analyze the impact of the distribution of the limited labeled data on the matching of single-cell data.

---

### Official Review · Reviewer_nJMm · 2024-11-02

**Soundness:** 2
**Presentation:** 3
**Contribution:** 1
**Rating:** 3
**Confidence:** 4

**Summary:**

This paper proposes a semi-supervised single-cell multi-omics data integration method based on neighborhood matching in the embedding space of auto-encoders. The proposed method is evaluated on three multi-omics datasets and proved effective.

**Strengths:**

1. The proposed method leverages several state-of-the-art machine learning techniques, achieving enhanced integration performance.
2. Ablation studies and parameter analyses are conducted to show the effectiveness and stableness of the proposed method.

**Weaknesses:**

1. Is the ZINB modeling suitable for scATAC-seq data? Could you provide any evidence that scATAC-seq data fits the ZINB distribution?
2. What is the difference between Eq. 5 and commonly used classification cross-entropy? Can authors provide some explanations and ablation studies on that?
3. Is the problem setting "realistic" as the authors claimed in the paper title? In what scenarios would we have part of annotated scRNA-seq data and part of annotated scATAC-seq data? In most cases, it seems more likely to have labeled scRNA-seq data and unlabelled scATAC-seq data. The setting in the evaluations seems to be even stranger. In real-world applications, when would we have and only have 50 to 200 labels?
4. The experimental comparisons are not fair. The proposed method requires paired multi-omics data as the input, in which the pairing information is a strong prior. In contrast, several baseline methods do not require the pairing information. The advantage of the proposed method could be attributed to the additional pairing and label information.
5. I note that Table 5 provides a more realistic data integration problem setting. However, the performance improvements of the proposed method are marginal, which limits the strength and significance of this work.
6. What is the time complexity of the proposed method? Does it scale to large datasets, given that it requires building the neighborhood graph?
7. I suggest the authors include baseline methods such as GLUE and MOFA+ in Table 1. The classification results could be obtained by using KNN on the integrated embeddings.

**Questions:**

I expect the authors to address my concerns raised in the weakness section. Currently, the paper seems to be a combination of several state-of-the-art machine learning methods, and the problem setting is not ''realistic'' as the authors claimed. The improvements over existing methods are also limited. These important points need to be clarified.

---

### Official Review · Reviewer_kf7r · 2024-11-05

**Soundness:** 3
**Presentation:** 2
**Contribution:** 2
**Rating:** 5
**Confidence:** 3

**Summary:**

This paper propose a novel multimodal single-cell matching method via mapping both multimodal data into a shared embedding space. The method constructs a geometric graph to indicate cross-modal relations between samples and to further explore the high-order relations. In addition, it is designed to leverage both labeled and unlabeled multimodal data by generating label distribution for unlabeled data. The empirical study shows the effectiveness of the proposed method.

**Strengths:**

1. GRACE proposes several techniques from different perspectives to solve the multimodal single-cell matching challenge, including joint representation learning, semantic alignment and nonparametric semi-supervised learning.
2. The empirical study demonstrates the effectiveness of the proposed GRACE and the raionality of each component.

**Weaknesses:**

1. The writing could be further improved. The notations in the current manuscript is a bit complex and unclear. Also, the notations in Figure 1 are gibberish.
2. The overall design of the framework is novel but somehow intricate, and thus the compele training process of the method is not very clear and needs to be further elaborated . In addition, the motivations of some design are not very convincing, such as the usage of ZINB distribution and consistency learning.
3. There are several hyperparameters in the propsoed method, making the method less practical in the real-world senarios.

**Questions:**

1. Please further describe the overall training process of the proposed framework.
2. Could you provide a model compexity analyasis for GRACE?
3. Why does GRACE choose to use ZINB distribution and how to calculate NB (equation 3) given x is a cell embedding?

---

### Note · Authors · 2024-11-15

I have read and agree with the venue's withdrawal policy on behalf of myself and my co-authors.